# The sensitivity of landscape evolution models to spatial and temporal rainfall resolution

T. J. Coulthard[1] and C. J. Skinner[1]

[1]Department of Geography, Environment and Earth Sciences, University of Hull, U.K.

*Correspondence to*: Tom Coulthard (T.Coulthard@hull.ac.uk)

**Abstract**

Climate is one of the main drivers for landscape evolution models (LEMs), yet its representation is often basic with values averaged over long time periods and frequently lumped to the same value for the whole basin. Clearly, this hides the heterogeneity of precipitation – but what impact does this averaging have on erosion and deposition, topography and the

final shape of LEM landscapes? This paper presents results from the first systematic investigation into how the spatial and temporal resolution of precipitation affects LEM simulations of sediment yields and patterns of erosion and deposition. This is carried out by assessing the sensitivity of the CAESAR-Lisflood LEM to different spatial and temporal precipitation resolutions – as well as how this interacts with different size drainage basins over short and long time scales. A range of simulations were carried out varying rainfall from 0.25 hour - 5 km to 24 hour - Lump resolution over three different sized

basins for 30 year durations. Results showed that there was a sensitivity to temporal and spatial resolution, with the finest leading to > 100 % increases in basin sediment yields. To look at how these interactions manifested over longer time scales, several simulations were carried out to model a 1000 year period. These showed a systematic bias towards greater erosion in uplands and deposition in valley floors with the finest spatial and temporal resolution data. Further tests showed that this effect was due solely to the data resolution, not from orographic factors. Additional research indicated that these differences

in sediment yield could be accounted for by adding a compensation factor to the model sediment transport law. However, this resulted in notable differences in the topographies generated, especially in third order and higher streams. The implications of these findings are that uncalibrated past and present LEMs using lumped and time averaged climate inputs may be under-predicting basin sediment yields, as well as introducing spatial biases through under-predicting erosion in first order streams but over predicting erosion in second, third order streams and valley floor areas. Calibrated LEMs may give

correct sediment yields but patterns of erosion and deposition will be different and the calibration may not be correct for changing climates. This may have significant impacts on the modelled basin profile and shape from long time scale simulations.

# 1    Introduction

Landscape Evolution Models (LEMs) have been extensively developed to understand how Earth surface processes influence drainage basin dynamics and morphology. One of the important forcings of erosion and morphodynamic change in these models is climate – usually in the form of precipitation. However, all LEMs use some degree of spatial and temporal
averaging for their driving climate or precipitation data. Spatially, rainfall (or climate parameters) are usually lumped over the whole basin and changed together. This clearly removes the effects of spatial heterogeneity in the rainfall that may be caused by atmospheric factors (i.e. convective vs frontal) or due to topography (orographic effects). Temporally, there is always some form of averaging, whether decadal, annual, daily or hourly, that conceals heterogeneity in the precipitation input. However, the temporal resolution may be important with, for example, short intense periods of rainfall being capable
of generating flooding that would not occur if it were averaged over a longer time period. There are important practical reasons for using averaged or coarse spatial and temporal resolution precipitation data, including data availability, model parsimony and model efficiency. Using spatially lumped climate values makes models simpler to construct and coarse temporal resolutions can make them faster to run by enabling longer time steps. Furthermore, the availability of high temporal and spatial resolution precipitation data is often poor – especially if the quality and validity of the data is
considered. Finally, many LEM studies run over tens to thousands of years where it is impossible to generate or reconstruct suitable records of precipitation of a high resolution.

There has been a limited exploration of precipitation resolution impacts in previous LEM studies, Sólyom and Tucker, (2007) argued that over the long time scales that LEMs are often applied spatial effects will become less important. For
example, as the modelled period of a simulation increases the probability for separate convective rainfall cells hitting all parts of the basin increases. Additionally, Sólyom and Tucker (2007) suggest that temporal effects become less important when the basin is of sufficient size that hydrological travel times from the top to the bottom of the basin are greater than the duration of precipitation events.

As morphodynamic changes within basins are heavily associated with basin hydrology, further insights may be drawn from hydrological modelling studies - where the influence of the spatial and temporal resolution of rainfall inputs has been discussed for over three decades (Lobligeois et al., 2014).  In this literature, there is a general agreement that finer detail in the representation of spatial and temporal variability of rainfall in a hydrological model will improve the outputs, especially when observing hydrographs from a single event (Beven and Hornberger, 1982; Bronstert and Bárdossy, 2003; Finnerty et
al., 1997; Hearman and Hinz, 2007; Ogden and Julien, 1994; Wainwright, 2002; Wilson et al., 1979). Coarser resolutions will result in a long, low intensity prediction of the runoff, and finer resolutions result in shorter, higher intensity predictions (Hearman and Hinz, 2007). Several authors have suggested that finer spatial resolutions are required for smaller basin sizes (Andréassian et al., 2001; Gabellani et al., 2007; Lobligeois et al., 2014), as coarser spatial resolutions tend to incorporate a

greater proportion of rainfall that did not fall within the basin. For example, Gabellani et al., (2007) found that to achieve a model performance of less than 5 % RMSE in the discharge, the spatial resolution of the rainfall is required to be no more than 20 % that of the basin size, and the temporal resolution no more than 20 % of the basin time of concentration. However, Lobligeois et al., (2014) argued that the appropriate resolution depends on the scale of the basin, the characteristics of the

basin and the characteristics of individual rainfall events. Additionally, Krajewski et al., (1991) claimed that the temporal resolution had a greater impact than the spatial resolution.

It is important to also consider that the uncertainty within rainfall products increases with finer spatial and temporal resolutions, meaning that improved model performance is tempered by increased uncertainty surrounding the precipitation

data (Mcmillan et al., 2012). Therefore, to reduce the propagation of rainfall uncertainties through a hydrological model, the spatial and temporal resolutions are often aggregated to coarser scales, such as a basin-average areal value, and daily, decadal, monthly or annual totals. In such cases, Segond et al., (2007) suggested that a reliable basin-average value is sufficient, provided that there is not enough rainfall variability to overcome any dampening effects of the basin. This is supported by Lobligeois et al., (2014), where results from modelling 181 basins in France across a range of hydro-climatic

conditions with lumped and semi-distributed hydrological models showed that in almost every case the performance of lumped and semi-distributed models were very similar. Other studies have also observed that models utilising basin-averaged rainfall show similar performances to those utilising more detailed rainfall (Kouwen and Garland, 1989; Nicótina et al., 2008; Pessoa et al., 1993). The apparent lack of improved performance with finer spatial and temporal resolutions may also be explained by the ability to highly calibrate hydrological model outputs to field data.

In summary – the above studies show that for basin scale hydrological modelling, the temporal and spatial resolution of rainfall can have an impact on model outputs, but the model calibration process can account for this. It would seem sensible to apply this knowledge to LEMs, however, erosion and deposition processes within drainage basins do not linearly reflect the hydrology. For example, an LEM driven by hourly precipitation data (Coulthard et al., 2012b) has shown that erosion,

deposition and sediment yields responded exponentially to flood size. Tucker and Hancock (2010) also identified this sensitivity of erosion and deposition to discharge in their review of LEMs. They examined research considering the role of discharge variability through time on erosion and landscape development. (e.g. Lague et al., 2005; Molnar et al., 2006; Tucker and Bras, 2000) and illustrated how precipitation variability can have an equal or greater erosive impact than precipitation amount (Tucker and Bras, 2000). Additionally, Tucker and Hancock (2010) noted that erosion and sediment

transport rates will also tend to increase with greater flow variability – and flow variability may in turn be affected by the spatial and temporal resolution of precipitation. Work has been carried out to account for precipitation resolution in the SIBERIA LEM, where Willgoose and Riley (1998) and Willgoose (2005) used a scaling approach to calibrate 'effective parameters'. This involved measuring runoff and sediment erosion at a high temporal resolution in test plots and using this to calibrate a relationship to mean annual sediment transport thereby accounting for changes in sediment yield due to temporal

precipitation resolution, though only in terms of bulk basin sediment yields. In their study, the impact of finer temporal resolution precipitation data is clearly important as Willgoose (2005; page 84) stated *""It is possible for this temporal resolution error in the simulated erosion record to be as much as an order of magnitude in size".*

Modelling studies have also shown that geomorphic responses can be chaotic and highly variable with similar size flood events delivering highly different volumes of sediment and producing significantly different patterns of erosion and deposition (Castelltort and Van Den Driessche, 2003; Coulthard and Van De Wiel, 2007; Coulthard et al., 1998, 2005; Simpson and Castelltort, 2012; Van De Wiel and Coulthard, 2010). Furthermore, (Coulthard et al., 2013a) showed that the timing and size of the flood wave generated by a hydrodynamic flow model in LEMs had an important impact on sediment

yield. It is therefore reasonable to expect that whilst there is a relationship between hydrology and erosion and deposition – basin *morphodynamics* may have greater sensitivities to spatial and temporal resolutions of rainfall data.

An additional, yet important difference between hydrological and LEM studies are the metrics used for model assessment. Hydrological studies are frequently measured on the basin hydrograph – a spatially lumped metric of water delivered over

time at the basin outlet. Whereas, landscape evolution models are assessed on the spatial patterns of erosion and deposition such as basin shape or hypsometry (Hancock et al., 2016; Tucker and Hancock, 2010). Furthermore, over longer time scales spatial patterns or erosion and deposition become more important, as positive feedbacks lead to streams/gulleys incising, growing and increasing their basin area. Therefore, the basin outlet metric used by most of the previously mentioned hydrological studies will be hiding evolving spatial heterogeneities within the basin that over time are important for LEMs.

This raises the following questions:

     a.   How does the spatial and temporal resolution of precipitation/climate data impact upon basin erosion and deposition patterns, and ultimately longer term landscape evolution?

     b.   Can any differences in sediment yield and erosion and deposition patterns due to spatial and temporal precipitation

resolution be accounted for via adjustment or calibration of model parameters?

     c.   Are the metrics and methods commonly used for assessing the performance of basin hydrological models appropriate for LEM and morphodynamic models?

This paper will address these three questions by developing the CAESAR-Lisflood LEM (Coulthard et al., 2013a) to incorporate spatially variable rainfall and use it to test the impacts of spatial and temporal precipitation resolution on basin

geomorphology over a range of basin sizes.

## 2      Methods

### 2.1      CAESAR-Lisflood and model developments

CAESAR-Lisflood is a grid based LEM that uses a hydrological model to generate surface runoff, which is then routed using a separate scheme generating flow depths and velocities. These are then used to drive fluvial erosion over several grainsizes integrated within an active layer system. In addition, slope processes (mass movement and soil creep) are also simulated. Previous versions of CAESAR-Lisflood used a lumped hydrological model based on TOPMODEL driven by one precipitation time series for the whole basin. This study required the spatial and temporal resolution of the precipitation inputs to be altered, which led to some model adaptations. A detailed description of the revised hydrological components is provided below, but for elaboration on the hydraulic, fluvial erosion and slope model operation readers are referred to Coulthard et al., (2013a).

The hydrological model within CAESAR-Lisflood is based on an adaptation of TOPMODEL (Beven and Kirkby, 1979) based on an area lumped exponential store of water, where storage and release of water is controlled by the $m$ parameter. $m$ is responsible for controlling the rise and fall of the soil moisture deficit (Coulthard et al., 2002) and therefore influences the characteristics of the modelled flood hydrograph (Welsh et al., 2009). Higher values of $m$ increase soil moisture storage leading to lower flood peaks and a slower rate of decline of the recession limb of the hydrograph, and therefore represent a well-vegetated basin (Welsh et al., 2009). Conversely, lower values of $m$ represent more sparsely vegetated basins. If the local rainfall rate $r$ (m.h$^{-1}$) specified by an input file is greater than 0, the total surface and subsurface discharge ($Q_{tot}$ in m$^3$.s$^{-1}$); is calculated using Equation (1).

$$Q_{tot} = \frac{m}{T} \log \left( \frac{(r - j_t) + j_t \exp\left(\frac{rT}{m}\right)}{r} \right)$$

$$j_t = \frac{r}{\left( \frac{r - j_{t-1}}{j_{t-1}} \exp\left( \left( \frac{(0 - r)T}{m} \right) + 1 \right) \right)}$$

Here, $T$ = time (seconds); $j_t$ = soil moisture store; $j_{t-1}$ = soil moisture store from the previous iteration. If the local rainfall rate $r$ is zero (i.e. no precipitation during that iteration), equation (2) is used:

$$Q_{tot} = \frac{m}{T} \log \left( 1 + \left( \frac{j_t T}{m} \right) \right)$$

$$j_t = \frac{j_{t-1}}{1 + \left( \frac{j_{t-1} T}{m} \right)}$$

Equations 1 and 2 calculate a combined surface and subsurface discharge, and these are separated prior to runoff flow routing. This is done using a simple runoff threshold, which is a balance of the hydraulic conductivity of the soil ($K$), the slope ($S$) and the grid cell size that here is 50m ($Dx$) (Coulthard et al., 2002) (equation 3).

$$Threshold = KS(Dx)^2$$

The volume of water above this threshold, or above a user-defined minimum value ($Q_{min}$), is subsequently treated as surface runoff and routed using the hydraulic model.

CAESAR-Lisflood can already use precipitation data over a range of temporal resolutions and for most previous applications this resolution has been hourly, though the model has also been run with daily (Coulthard et al., 2013b) and at ten minute resolutions (Coulthard et al., 2012a). To enable spatially variable precipitation inputs and hydrology, CAESAR-Lisflood was modified so that precipitation rates could be input via spatially fixed pre-defined areas. These areas are defined with a

raster index file with numbers corresponding to the areas. In this study regular square areas of rainfall were used that corresponded with the available rainfall data, but any shape area can be used. For each area, a separate version of the hydrological model (equations 1-3) is run, enabling different levels of storage and runoff to be generated in different areas.

The volume of runoff in a cell (determined by the hydrological model above) is then treated as surface flow and routed

across the DEM surface using the Lisflood-FP hydrodynamic flow model developed by Bates et al., (2010) described further in Coulthard et al., (2013a). This model is a 2D hydrodynamic model containing a simple expression for inertia. Flow is routed to a cell's four Manhattan neighbours using Equation (4)

$$q_{t+\Delta t} = \frac{q_t - gh_t\Delta t \frac{\partial(h_t + z)}{\partial x}}{\left(1 + gh_t\Delta t \ ^2 q_t / h_t^{10/3}\right)}$$

where $\Delta t$ = length of time step (s); $t$ and $t + \Delta t$ respectively denote the present time step and the next time step; $q$ = flow per unit width (m$^2$.s$^{-1}$); $g$ = gravitational acceleration (m.s$^{-2}$); $h$ = flow depth (m); $z$ = bed elevation (m); $x$ = grid cell size (m); $\frac{\partial(h_t+z)}{\partial x}$ = water surface slope and $n$ is Manning roughness coefficient.

Flow depths and velocities determined by the hydraulic model are then used to calculate a shear stress that is then fed into a sediment transport function to model fluvial erosion and deposition. CAESAR-Lisflood provides a choice of sediment transport function with the Einstein (1950) or, as used in this study, the Wilcock and Crowe (2003) method. Sediment

transport is then determined for (up to) nine different grainsize classes and these may be transported as bedload or suspended load. A distinction is made between the deposition of bed load and suspended load, where bedload is moved directly from cell to cell, whereas fall velocities and the concentration of sediment in within a cell determine suspended load deposition. Importantly, the incorporation of multiple grainsizes, and model formulation of the selective erosion, transport and deposition of the different size allows a spatially variable sediment size distribution to be modelled. However, as this grainsize heterogeneity is expressed vertically as well as horizontally, a method for storing sub-surface sediment data is required. This is achieved by using a system of active layers comprising: a surface active layer (the stream bed); multiple buried layers (strata); and, if needed, an un-erodible bedrock layer (Van De Wiel et al., 2007). Slope processes are also simulated, with landslides occurring when a user defined slope threshold is exceeded and soil creep carried out as a linear function of slope.

## 2.2    Study area

The basin studied is the River Swale in Northern England. The mean basin relief is 357 m, ranging between 68-712 m with an average river gradient of 0.0064. The headwaters of the Swale are characterized by steep valleys and the geology is Carboniferous limestone and millstone grits (Bowes et al., 2003). Downstream, valleys are wider and less steep, with the underlying geology becoming Triassic mudstone and sandstones (Bowes et al., 2003). This basin has been extensively modelled in previous studies (Coulthard and Macklin, 2001; Coulthard and Van de Wiel, 2013; Coulthard et al., 2012b, 2013a), and a pre-calibrated version of the CAESAR-Lisflood model was readily available. The basin was sub-divided to provide test sub basins of various sizes giving three basin sizes – herein referred to as the Complete Swale, the Upper Swale and the Arkengarthdale tributary (Figure 1; Table 1).

## 2.3    Precipitation data, and model configuration

The rainfall data used was derived from the UK Composite NIMROD rainfall radar (Met Office, 2003) that has a native resolution of 5 km grid cells and 0.25 hour time steps with rainfall intensities in mm.hr$^{-1}$. The maximum available ten-year record was extracted from the period 2004 - 2014 for the 5 km cells lying over the Swale basin. The 0.25 hour - 5 km resolution were the finest scale data available and to provide a range of different resolutions these data were re-sampled to the scales detailed below (Table 2). When aggregating to coarser spatial scales, the relative contribution of each 5 km cell was weighted, equal to its relative contribution to the basin. Therefore, with each spatial resolution the same volume rain input is applied at each daily time step. Note, that this differs from some studies of the effect of spatial resolution where some of the variation can be explained by producing rainfall records from a domain that exceeds the bounds of the basin. Here, only the spatial representation of the same rainfall record was examined.

To study the absolute role of spatial and temporal precipitation data, a matrix of model runs was carried out as shown in Table 2. Spatially, rainfall was lumped into basin-average (henceforth "Lump"), 20, 10 and 5 km cells, and temporally

averaged into 24, 12, 8, 6, 4, 1 and 0.25 hour time steps. This matrix of runs was applied to all three basins (Complete Swale, Upper Swale and Arkengarthdale) though as the smallest basin (Arkengarthdale) is smaller than the 20 km resolution grid cell, it is run at 5 km, 10 km and Lump spatial resolutions.

To investigate any longer term impacts of precipitation resolution, two 1000 year simulations were carried out on the Upper Swale basin using the end members of our driving data, the 24 hour - Lump and 0.25 hour - 5 km resolution data. Both simulations used the 10 year precipitation record looped 100 times.  However, as this test would result in the same spatial patterns of rainfall being applied to the same area one hundred times that could bias areas of erosion and deposition to those receiving the most precipitation – rather than being a test of the spatial and temporal resolution. Therefore, to disrupt any

spatial pattern legacies, two additional 1000 year simulations were carried out where after every simulated ten years, the locations of each rainfall pixels were randomly reassigned (called random 1 and random 2). Only two random simulations were carried out, due to long model run times and as these random runs gave very similar results. Results from the two long term simulations were then compared against both of these random simulations.

One aim of this research is to see whether any changes in sediment yields and erosion and deposition patterns due to the spatial or temporal resolution of the precipitation could be accounted for by adjusting model parameters. To investigate this three series of comparison runs were carried out – where a factor was added to the sediment transport model to allow erosion totals to be adjusted to match. This allowed us to normalise the sediment yields from the simulations being compared, with the aim of observing any differences in spatial patterns of erosion and deposition in the DEMs. The following three sets of

comparisons were carried out: to compare patterns of erosion and deposition for spatial resolution changes (0.25 hour - Lump vs 0.25 hour - 5 km); temporal resolution changes (24 hour - Lump vs 0.25 hour - Lump); and spatial and temporal resolutions (24 hour - Lump vs 0.25 hour - 5 km).

In order to adjust the sediment output, an additional term or compensation factor $C$ was added to the Wilcock and Crowe,

(2003) sediment transport formula used here, as shown in equation 5 below (as fully described in Van De Wiel et al., (2007) . Here, $q_i$ is the sediment transport rate in $m^3 s^{-1}$, $g$ is gravitational acceleration ($m\ s^{-2}$), $\rho_s$ and $\rho$ are the density of sediment and water respectively, $F_i$ is the fractional volume of the $i$-th sediment in the top active layer, $U_*$ is the shear velocity ($U_* = [\tau/\rho]^{0.5}$) and $W_i^*$ is a function that relates the fractional transport rate to the total transport rate.

$$q_i = C \frac{F_i U_*^3 W_i^*}{((\rho_s - \rho) - 1)g}$$

For each of the three comparisons identified above (e.g. 24 hour - Lump, 0.25 hour - 5 km and 0.25 hour - Lump) runs were carried out varying $C$ from 1 to 2.5 in increments of 0.1, resulting in an additional thirty 1000 year simulations. After this,

the closest matching sediment yields over the 1000 years were used to compare differences in spatial patterns of erosion and deposition.

A final test, was to determine whether or changes in erosion and deposition were due to orographic effects. Previous research indicates a geomorphic sensitivity in the CAESAR model to rainfall magnitudes (Coulthard et al., 2012b) so it is therefore important to disentangle whether any increased erosion totals were due to the precipitation data resolution or orography. To test for this we carried out a series of additional simulations using the 0.25 hour - 5 km data where the 5 km rainfall grid cells were randomly re-distributed or 'jumbled' to produce 20 different records. These jumbled data were then averaged to each of the temporal resolutions and the 30 year simulations were re-run.

All simulations were carried out over a 50 m resolution DEM, with no bedrock representation and no vegetation parameters applied. Excluding the 1000 year simulations, all runs shown in Table 2 were carried out for a 30 year period based on the 10 year rainfall record repeated three times. The initial simulation conditions were based on a pre-conditioned 'spun up' DEM generated by a 30 year model run using the 24 hour - Lump rainfall. This 'spinning up' process removes sharp gradient changes in the DEM that may be a legacy of its generation and also allows the model to evolve a surface channel grainsize distribution from the initial global distribution described in Table 3. Apart from rainfall parameters and the basin DEM, all model parameters were kept constant, with one exception where the input/output difference allowed was set at 10 $m^3.s^{-1}$ for the Complete Swale, 5 $m^3.s^{-1}$ for the Upper Swale and 2.5 $m^3.s^{-1}$ for the Arkengarthdale. This ensured that the model ran efficiently, with each value appropriate for the respectively basin size, and the hydrological regime. A list of CAESAR-Lisflood parameter values used in the simulations is shown in Table 3. The 1000 year simulations were carried out for the upper Swale only, as the longest run times here were 4 weeks, compared to 8+ weeks for the whole Swale.

For all simulations, water and sediment outputs were sampled from the model at 0.25 hour time steps and the DEM saved every 10 simulated years. From these data, mean annual output and values above the 95th percentile (representing peaks) were calculated for both water (discharge rate $m^3.s^{-1}$) and sediment yield ($m^3$). To allow better comparison between the basin sizes, the above metrics were calculated as a percentage deviation from the baseline, which was taken to be the 24 hour - Lump resolution. To assess the impact of different resolutions on the modelled basin hydrology, outputs were compared to discharge data at Catterick Bridge using RMSE and Nash-Sutcliffe metrics for the ten year record 2004 - 2014.

## 3      Results

### 3.1      Hydrology

Table 4 shows the influence that the spatial and temporal resolutions of the rainfall data have on the peak rainfall intensities, showing a marked increase with finer resolutions, as found in previous hydrological studies (e.g. Tustison et al., 2001).

These changes are largely translated through to the basin hydrology (Tables 5 and 6) but with some differences. Looking at mean annual discharge (Table 5) there is an increase in water output with finer temporal and spatial resolution for the largest Complete Swale basin, though only an increase with finer temporal resolution for the smaller two basins. Looking at peak values (Table 6) these changes are even less apparent with most differences linked to finer temporal resolutions. However, overall these changes are relatively minor (c.4 % for mean annual discharge and 5-10 % for peak discharges) especially when compared to the difference in maximum rainfall intensity (Table 4).

A full evaluation of hydrological performance could not be ascertained because the location of the nearest gauging station incorporates the contribution of a tributary outside of the modelled domain, though as this tributary was small in comparison to the whole domain a relative comparison could still be made. The relative comparisons showed that the change in resolution also influences the performance statistics for the hydrology with Table 7 showing an improvement in performance (RMSE and Nash-Sutcliffe) with finer temporal resolution with only very small improvements due to finer spatial resolutions (RMSE only).

### 3.2     Sediment outputs

Tables 8 and 9 describe how with changing temporal and spatial resolution there is a clear trend of increasing sediment yields with finer spatial and temporal resolutions. Compared to basin hydrology, the results show that the sediment yield is notably more sensitive, with the greatest deviation being 118.1 % in the mean annual volumes, with the corresponding hydrological deviation being 2.8 %. Each basin shows a sensitivity to spatial resolutions, which increases with the basin size though differences are reduced between the 1 hour and 0.25 hour temporal resolutions.

For the 1000 year simulations, there are differences in erosion and deposition patterns between the random 1 (with 0.25 hour - 5 km resolution data) and the 24 hour - Lump simulation (Figure 2). Notably there is more erosion in all headwater and first order streams and substantial amounts of deposition in the valley floors. The six cross sections (Figure 3, A to F) provide more detail on morphological changes at these sites with 3-5 m additional incision at cross section B and 6m at cross section D, along with up to 3 m of deposition at cross sections D and E. Interestingly these are not restricted to single channel threads, at E this occurs across some 350 m of valley floor. The results for the random 1 simulation were very similar to those for the random 2 simulation (not shown). However, there was a significant difference between random 1 and the 1000 year 0.25 hour -5km resolution simulation and the random runs. These differences are a facet of repeating the 10 year rainfall sequence 100 times and are presented in Figure 5, where the most notable difference is > 2.5 m in the valley floor to the Western side of the basin along with smaller changes in the valley floor downstream.

For the 1000 year comparison runs, to account for the *temporal* resolution difference, the 24 hour - Lump simulations gave sediment yields that were within 3 % of the 0.25 hour - Lump simulation with a compensation factor of 2.0. For *spatial*, 0.25

hour - Lump with a compensation factor of 1.1 gave yields within 4 % of 0.25 hour - 5 km simulations and for *spatial and temporal*, 24 hour - Lump with a compensation factor of 2.2 was within 5 % of the 0.25 hour - 5 km simulation. Figures 6, 7 and 8 describe the spatial patterns of the differences between the two final DEMs from these simulations. These show areas of greater difference in the lower half of the basin for different temporal resolution data, in the upper half of the basin for spatial resolution data and in both upper and lower for changes in spatial and temporal resolution. In each Figure a series of cross sections are highlighted to illustrate vertical changes.

For the simulations to test the impact of orography, Figure 9 shows there is a general, though not significant orographic relationship for rainfall intensity in the Swale. However, the results of the jumbled runs (Figure 10) show that as the temporal resolution of the rainfall increases, so does the hydrological and sediment totals from each run. Furthermore, the trend lines show a clear offset between the different resolutions. This strongly indicates that it is the spatial and temporal resolution and not any orographic effects within the data that are responsible for increased sediment yields described previously.

## 4        Discussion

### 4.1        The impact of precipitation spatial and temporal resolution on sediment yield and longer term landscape evolution

Clearly both temporal and spatial resolution of precipitation has an important effect on the amount of sediment coming from a basin, and where it is eroded and deposited (Tables 8 and 9; Figures 2 and 3). Finer resolution (spatial and temporal) rainfall inputs can increase the local rainfall intensity over parts of the basin, whilst the overall rainfall volume remains the same. This leads to a slight increase in the basin water discharge (< 5 %), but a much greater increase in sediment yields - that are in some cases doubled (Tables 8 and 9). Changes in rainfall resolution also alter spatial patterns of erosion and deposition (Figures 2 and 3) with increased local rainfall intensities (from finer resolution data) leading to increased local runoff and thus increased erosion in the smaller first and second order streams (Figure 2, cross sections B, D, F). The transporting capacity of third order streams, however, does not increase proportionally to the increased sediment supply from the first and second order streams as the peak discharges in the larger systems are less sensitive to locally high rainfall intensities, thereby limiting erosion in third order streams. The Lump simulations lead to lower erosion rates in the first and second order streams so that transporting capacity is not reached in the third order streams, leading to greater amounts of erosion (and less deposition) in third order streams and valley floor sections (e.g. Figure 3, cross sections A and E). The disproportionate relationship between changes in hydrology and erosion/deposition is highly important in this context, as small changes in hydrology (here local and temporal) can clearly have a significant impact on basin sediment yield and local erosion/deposition patterns. This affirms the findings of (Coulthard et al., 2012b) where they noted the 'geomorphic multiplier' effect between rain, runoff and sediment yield.

For existing and previous LEM studies these results suggest that there may have been a systematic under-representation of basin wide sediment yields by using lumped and coarse temporal resolution climate/precipitation data. These is may not be of concern to many LEM studies, that are interested in exploring general relationships between processes, drivers and subsequent landscape change. However, the spatial changes in erosion and deposition patterns generated by the different resolution rainfall data will affect results and findings. Over the 1000 years we have simulated in this study, the coarser resolution data leads to more incision/erosion in third order and higher streams and less in first and second order streams. This has led to a change in the shape of the basin long profile - and thus when projected over even longer time scales will lead to changes in the shapes of predicted basins, landscapes and landforms. Resolving this is troublesome as for many existing models, especially those dealing with longer time scale simulations (e.g. > 10 000 years), incorporating high resolution precipitation data is impractical. The data is simply not available and generating synthetic rainfall complex. Therefore, can these changes in erosion and deposition patterns (and sediment yields) be compensated for via model adjustment rather than calibration?

## 4.2    Adjusting to compensate for spatial and temporal rainfall resolution effects

In our final set of 1000 year comparison runs, erosion and deposition totals were adjusted so simulations with different spatial and temporal rainfall resolutions could be compared. Sediment yields could easily be matched, however there were differences in erosion and deposition patterns produced by different temporal (Figure 6) and spatial resolutions (Figure 7). This indicates that such adjustment (similar to that carried out by Willgoose and Riley, 1998) can be carried out, but with some notable effects and possible limitations.

Adjusting for temporal changes in rainfall resolution led to good results in the upland, Western side of the basin, with very few areas where there were differences in erosion and deposition patterns greater than 0.5 m (Figure 6, cross section A). However, changes in the valley floor sections lower down are greater, with more erosion/less deposition generated by the adjusted 24 hour resolution simulation (Figure 6 cross sections B and C). In the steeper upland areas a larger compensation factor (2.0) is required for the 24 hour rainfall to generate similar amounts of erosion as the more intense, flashier events from the 0.25 hour data. As the 2.0 compensation factor is applied globally, in the lower parts of the basin where there is less difference in flow magnitudes generated by the 0.25 and 24 hour runs, this leads to more erosion or less deposition.

Conversely, adjusting for spatial resolution leads to very small differences in the lower, Eastern sections (Figure 7, cross sections B and C) but major changes in the upland area with 8 m more erosion/less deposition from the adjusted lump simulation at Figure 7 cross section A. As for Figure 2, by lumping local rainfall heterogeneity there are smaller flows in first and second order streams, but greater in third – leading to the incision at cross section A. Here the incision at A is

greater than values shown in Figure 2 as the temporal resolution of the data is 0.25 hour rather than 24 hour. There are very few differences in the lower sections of the basin (Figure 7, cross sections B and C) as for both simulations here the flows will be the cumulative of the total basin rainfall (as rain cells are reassigned every 10 years in the 0.25 hour - 5 km random simulation).

Adjusting for both (Figure 8) best represents how most LEMs may be adjusted for using coarser spatial and temporal resolution data. This comparison required our largest compensation factor (2.2) and generated patterns that could be described as a merging of Figures 6 and 7 – with more erosion/less deposition in cross sections A, B and C. As per the discussion for Figure 2, the coarser resolution data drives more erosion/less deposition in third order streams and valley floor

10 areas – if continued for more thousands of years this would result in a considerably different long profile, which would change the morphometry of the basin.

There are further difficulties associated with adjusting models to compensate for different resolution data. For example, if we have a convective 4 hour event of 6 mm.hr$^{-1}$ and a synoptic 24 hour event of 2mm.hr$^{-1}$ then adjusting erosion rates for 24

15 hour resolution data would scale erosion from the convective storm down and the synoptic event up. This adjustment, however, would assume the same erosion/deposition relationship between our synoptic and convective event and in the context of climate change it is highly likely that this will change. For example, climate changes may lead to more or less rainfall as well as greater or lesser rainfall durations and intensities. In other words, the relationship between mean annual erosion rates and mean annual rainfall is non-stationary, yet any calibrated adjustment or scaling factor is fixed to the

20 properties of the period it was calibrated against. This could readily lead to "over-calibration", a phenomena noted by the hydrological community (Andréassian et al., 2012), where the parameters in hydrological models are adjusted too tightly based on too few observations. The issue of non-stationary calibration of parameters is also widely acknowledged in the hydrological modelling literature (for example, Beven, 2006) where the period simulated is far shorter, and therefore possibly less varied, than the longer time scales over which LEMs may operate.

In summary – the adjustment of model parameters can be used to compensate basin sediment yields for different resolution rainfall data, but there is an impact on patterns of erosion and deposition within the basin. Using such adjustments is likely to be basin specific and the correction will not be correct over changing climates. Calibration, the adjustment of sediment yields to match field data, will likely encounter the same issues.

### 4.3 Are hydrological basin wide metrics suitable for LEM/Morphodynamic models?

This study raises some interesting issues regarding the suitability of hydrological type metrics (e.g. basin discharge) for evaluating LEMs, morphodynamic or geomorphic models. Basin sediment yield may be a useful indicator of overall LEM

performance, but will conceal much of the important geomorphic change within a basin. Therefore, a good hydrological and/or sediment yield prediction from a LEM does not necessarily translate to a good morphodynamic prediction. Similar hydrographs of water and sediment at a basin exit may come from completely different parts – and leave a very different geomorphic signature. Here is an important distinction between the hydrology and geomorphology – as different hydrological responses will not necessarily leave any sort of hydrological record in the system. But geomorphological changes in response to the hydrology will.

Largely model metrics are driven by the aims of the model. For example, a hydrograph may be a very useful output for a basin hydrological model (to feed in, for example, into a flood model). Whereas for a morphodynamic model we are interested in the changes occurring throughout the basin not just those reflected at the end. This is especially important for LEMs where patterns of erosion and deposition feedback to control the shape of basins and landscape development – and this effect increases with the duration of model study or simulation. This point is identified in recent work by (Hancock et al., 2016) showing that using the SIBERIA model, over 10 000 years different shape landscapes can evolve yet generate very similar sediment yields.

## 4.4    Limitations

It is important to consider that these findings are based on numerical simulations that contain many of simplifications and assumptions. CAESAR-Lisflood is driven by a hydrological model where changes in land-use are represented through altering model parameters ($m$) leading to flashier or more reduced hydrographs. This may prove to be a considerable sensitivity to precipitation temporal and spatial resolution and in these simulations we have deliberately used a moderate value for an $m$ of 0.01 – which in previous CAESAR-Lisflood simulations has been used to represent natural scrubland. We would suggest that lower values for grassland (e.g. 0.005) would increase sensitivity and larger for forest/woodland (0.02) would reduce sensitivity, though further simulations would be required to show this. Basin hydrology is a balance between precipitation, evaporation, infiltration and groundwater effects. These processes are all spatially and temporally variable but we have quite deliberately only altered the rainfall to determine model sensitivity to just this parameter. Within CAESAR-Lisflood the TOPMODEL $m$ parameter is used to account for evaporation, infiltration and groundwater effects and can also be changed spatially and temporally (Coulthard and Van De Wiel, 2016). Examining model sensitivity to both may be useful future research.

There may also be issues with the DEM resolution (here 50 m) and how that interacts with different spatial resolutions of rainfall inputs with other workers showing that grid resolution in LEMs can have an impact (Hancock et al., 2016). Furthermore, there are uncertainties associated with the upscaling of the precipitation data and the transfer of rain radar data to actual values. However, notwithstanding the above limitations, our results provide very useful insight into how spatially and temporally changing precipitation can alter simulated basin geomorphology and sediment yields.

## 5    Conclusions

Our findings show that simulated basin sediment yields and spatial patterns of erosion and deposition are sensitive to the spatial and temporal resolutions of precipitation data used to drive models. The impact of temporal changes is greater than that of spatial changes. Using finer resolution data for both leads to significant increases in sediment outputs, with 0.25 hour - 5 km resolution data leading to a doubling in basin sediment yields over with the 24 hour - Lump data. These changes are due to finer resolution data generating increased erosion in upland and first order streams with increased deposition and aggradation in valley floor areas. Further simulations indicated that the differences in total sediment yield could be removed with a compensation/adjustment factor inserted in the sediment transport law. However, using such a factor resulted in notable differences in the topographies generated, especially in third order and higher streams. Overall, the implications of these findings are that uncalibrated past and present LEMs using coarse spatial and temporal resolution precipitation drivers may be under-predicting basin sediment yields, under-predicting erosion in first order streams but over predicting erosion in third order streams and valley floor areas. Calibrated LEMs may give correct sediment yields but patterns of erosion and deposition will be different and the calibration may not be correct for changing climates. It is highly likely this will have significant impacts on the modelled basin profile and shape from long time scale simulations. Our findings are placed in the context of LEMs – but it should be considered that such issues of rainfall spatial and temporal resolution may be highly important to soil erosion models, and other basin based sediment models that may be using coarser resolution precipitation data.

**Acknowledgements**

The Authors would like to thank the Editors, the two reviewers and Declan Valters for their comments and suggestions, all of which contributed to greatly improving this manuscript. This work was funded by the NERC Flash Flooding from Intense Rainfall (FFIR) funded project, Susceptibility of Basins to Intense Rainfall and Flooding (SINATRA) NE/K008668/1.

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

*Table 1. Basin areas and elevations for the three test basins used.*

| Catchment | Area (km$^3$) | Minimum Elevation (m) | Maximum Elevation (m) |
|---|---|---|---|
| **Complete Swale** | 415 | 68 | 712 |
| **Upper Swale** | 181 | 182 | 712 |
| **Arkengarthdale** | 62 | 198 | 664 |

*Table 2 Matrix of runs using different temporal (x) and spatial (y) resolutions.*

| | **24 Hour** | **12 Hour** | **8 Hour** | **4 Hour** | **1 Hour** | **0.25 Hour** |
|---|---|---|---|---|---|---|
| **Lump** | 24 Hour - Lump | 12 Hour - Lump | 8 Hour - Lump | 4 Hour - Lump | 1 Hour - Lump | 0.25 Hour - Lump |
| **20 km** | 24 Hour- 20 km | 12 Hour- 20 km | 8 Hour - 20 km | 4 Hour - 20 km | 1 Hour - 20 km | 0.25 Hour - 20 km |
| **10 km** | 24 Hour- 10 km | 12 Hour- 10 km | 8 Hour - 10 km | 4 Hour - 10 km | 1 Hour - 10 km | 0.25 Hour - 10 km |
| **5 km** | 24 Hour- 5 km | 12 Hour- 5 km | 8 Hour - 5 km | 4 Hour - 5 km | 1 Hour - 5 km | 0.25 Hour - 5 km |

*Table 3. CAESAR-Lisflood model parameters used.*

| CAESAR-Lisflood Parameter | Values |
|---|---|
| Grainsizes (m) | 0.0005, 0.001, 0.002, 0.004, 0.008, 0.016, 0.032, 0.064, 0.128 |
| Grainsize proportions (total 1) | 0.144, 0.022, 0.019, 0.029, 0.068, 0.146, 0.220, 0.231, 0.121 |
| Sediment transport law | Wilcock & Crowe |
| Max erode limit (m) | 0.002 |
| Active layer thickness (m) | 0.01 |
| Lateral erosion rate | 0.0000005 |
| Lateral edge smoothing passes | 40 |
| m value | 0.01 |
| Soil creep/diffusion value | 0.0025 |
| Slope failure threshold | 45 degrees |
| Evaporation rate (m/day) | 0 |
| Courant number | 0.7 |
| Mannings n | 0.04 |
|  |  |

*Table 4. Maximum rainfall intensities from the ten year record for each resolution, taken from the domain for the Complete Swale catchment.*

| Maximum Rate (mm.hr$^{-1}$) | 24 hour | 12 hour | 8 hour | 6 hour | 4 hour | 1 hour | 0.25 hour |
|---|---|---|---|---|---|---|---|
| Lump | 2.87 | 4.08 | 5.90 | 5.96 | 7.83 | 17.30 | 37.74 |
| 20 km | 3.29 | 5.03 | 7.10 | 8.21 | 10.54 | 18.66 | 70.63 |
| 10 km | 3.29 | 5.03 | 7.10 | 8.21 | 10.54 | 19.06 | 70.63 |
| 5 km | 4.06 | 5.77 | 7.58 | 8.70 | 11.24 | 25.23 | 76.75 |

*Table 5. The percentage deviations of the mean annual hydrological outputs using different spatio-temporal resolutions, for each catchment.*

| | 24 hour | 12 hour | 8 hour | 6 hour | 4 hour | 1 hour | 0.25 hour |
|---|---|---|---|---|---|---|---|
| **Complete Swale** | | | | | | | |
| Lump | 0.00 | 1.19 | 1.61 | 1.54 | 1.68 | 1.63 | 1.66 |
| 20 km | 0.80 | 1.62 | 1.90 | 2.11 | 2.36 | 2.53 | 2.49 |
| 10 km | 0.74 | 1.72 | 2.15 | 2.38 | 2.55 | 2.58 | 2.61 |
| 5 km | 0.76 | 1.96 | 2.35 | 2.52 | 2.68 | 2.81 | 2.82 |
| **Upper Swale** | | | | | | | |
| Lump | 0.00 | 1.05 | 1.40 | 1.61 | 1.71 | 1.90 | 1.97 |
| 20 km | -0.08 | 0.93 | 1.38 | 1.50 | 1.74 | 1.88 | 1.91 |
| 10 km | 0.21 | 0.96 | 1.57 | 1.65 | 1.81 | 2.00 | 2.05 |
| 5 km | 0.22 | 1.13 | 1.69 | 1.67 | 1.85 | 2.01 | 2.00 |
| **Arkengarthdale** | | | | | | | |
| Lump | 0.00 | 2.27 | 2.88 | 3.26 | 3.76 | 4.33 | 4.32 |
| 10 km | -0.78 | 2.28 | 2.67 | 3.12 | 3.74 | 4.27 | 4.26 |
| 5 km | -0.94 | 2.26 | 2.26 | 3.07 | 3.44 | 4.21 | 4.29 |

*Table 6. The percentage deviations of the volume of hydrological outputs above the 95th percentile using different spatio-temporal resolutions, for each catchment.*

| | 24 hour | 12 hour | 8 hour | 6 hour | 4 hour | 1 hour | 0.25 hour |
|---|---|---|---|---|---|---|---|
| **Complete Swale** | | | | | | | |
| Lump | 0.00 | 3.72 | 4.19 | 4.65 | 4.96 | 5.16 | 5.15 |
| 20 km | 0.32 | 4.05 | 4.53 | 5.14 | 5.50 | 5.76 | 5.75 |
| 10 km | 0.46 | 4.25 | 4.76 | 5.38 | 5.74 | 5.99 | 6.00 |
| 5 km | 0.16 | 3.96 | 4.49 | 5.16 | 5.51 | 5.72 | 5.75 |
| **Upper Swale** | | | | | | | |
| Lump | 0.00 | 3.61 | 4.82 | 5.27 | 5.58 | 6.05 | 6.08 |
| 20 km | -0.06 | 3.51 | 4.69 | 5.14 | 5.45 | 5.89 | 5.93 |
| 10 km | -0.02 | 3.58 | 4.78 | 5.25 | 5.57 | 6.05 | 6.09 |
| 5 km | -0.24 | 3.41 | 4.47 | 4.97 | 5.31 | 5.77 | 5.72 |
| **Arkengarthdale** | | | | | | | |
| Lump | 0.00 | 6.75 | 7.26 | 8.33 | 8.94 | 9.64 | 9.78 |
| 10 km | -0.05 | 8.38 | 7.27 | 8.31 | 8.89 | 9.56 | 9.70 |
| 5 km | -0.12 | 6.56 | 7.15 | 8.35 | 8.86 | 9.64 | 9.70 |

*Table 7. Hydrological performance statistics from the Upper Swale catchment, comparing daily discharges from the CAESAR-Lisflood model and observed daily discharges recorded from Catterick Bridge. Red shading indicates the worst performance statistics, and the green the best performance statistics.*

| RMSE (m³.s⁻¹) | 24 hour | 12 hour | 8 hour | 6 hour | 4 hour | 1 hour | 0.25 hour |
|---|---|---|---|---|---|---|---|
| Lump | 20.58 | 19.37 | 18.61 | 18.05 | 17.54 | 16.72 | 16.50 |
| 20 km | 20.59 | 19.46 | 18.68 | 18.13 | 17.61 | 16.72 | 16.52 |
| 10 km | 20.57 | 19.47 | 18.69 | 18.14 | 17.59 | 16.70 | 16.50 |
| 5 km | 20.55 | 19.50 | 18.74 | 18.19 | 17.64 | 16.74 | 16.53 |

| Nash-Sutcliffe | 24 hour | 12 hour | 8 hour | 6 hour | 4 hour | 1 hour | 0.25 hour |
|---|---|---|---|---|---|---|---|
| Lump | 0.24 | 0.33 | 0.38 | 0.42 | 0.45 | 0.50 | 0.51 |
| 20 km | 0.24 | 0.32 | 0.38 | 0.41 | 0.45 | 0.50 | 0.51 |
| 10 km | 0.24 | 0.32 | 0.38 | 0.41 | 0.45 | 0.50 | 0.51 |
| 5 km | 0.25 | 0.32 | 0.37 | 0.41 | 0.45 | 0.50 | 0.51 |

*Table 8. The percentage deviations of the mean annual sediment yield outputs using different spatio-temporal resolutions, for each catchment.*

| Complete Swale | 24 hour | 12 hour | 8 hour | 6 hour | 4 hour | 1 hour | 0.25 hour |
|---|---|---|---|---|---|---|---|
| Lump | 0.00 | 44.04 | 51.96 | 48.54 | 53.50 | 66.50 | 66.18 |
| 20 km | 27.78 | 63.16 | 72.56 | 73.12 | 83.15 | 91.09 | 91.74 |
| 10 km | 30.99 | 64.85 | 78.46 | 72.59 | 87.91 | 98.71 | 100.54 |
| 5 km | 34.72 | 67.94 | 90.64 | 84.03 | 101.28 | 115.00 | 118.10 |
| **Upper Swale** | | | | | | | |
| Lump | 0.00 | 16.14 | 22.77 | 22.39 | 29.88 | 35.02 | 40.25 |
| 20 km | -2.45 | 14.18 | 15.28 | 20.36 | 26.06 | 34.00 | 37.49 |
| 10 km | -4.19 | 14.68 | 20.45 | 23.21 | 28.81 | 38.02 | 38.85 |
| 5 km | 3.02 | 22.70 | 29.75 | 37.81 | 41.30 | 52.93 | 52.56 |
| **Arkengarthdale** | | | | | | | |
| Lump | 0.00 | 30.06 | 42.76 | 54.01 | 58.83 | 75.95 | 77.44 |
| 10 km | -1.15 | 37.84 | 49.28 | 53.23 | 61.45 | 75.01 | 74.75 |
| 5 km | -4.20 | 50.49 | 50.49 | 61.63 | 67.36 | 87.34 | 80.74 |

*Table 9. The percentage deviations of the volume of sediment yield outputs above the 95th percentile using different spatio-temporal resolutions, for each catchment.*

| Complete Swale | 24 hour | 12 hour | 8 hour | 6 hour | 4 hour | 1 hour | 0.25 hour |
|---|---|---|---|---|---|---|---|
| Lump | 0.00 | 44.54 | 49.62 | 48.47 | 54.83 | 63.76 | 63.50 |
| 20 km | 17.81 | 53.18 | 66.84 | 62.02 | 72.51 | 79.50 | 82.67 |
| 10 km | 23.26 | 51.26 | 69.28 | 56.03 | 72.42 | 84.76 | 84.67 |
| 5 km | 25.28 | 54.26 | 78.76 | 70.22 | 85.10 | 96.84 | 99.21 |
| **Upper Swale** | | | | | | | |
| Lump | 0.00 | 20.08 | 26.65 | 27.70 | 34.05 | 39.88 | 43.84 |
| 20 km | -2.57 | 18.03 | 20.03 | 24.82 | 30.96 | 38.02 | 40.99 |
| 10 km | -3.85 | 17.94 | 23.75 | 26.99 | 32.98 | 41.57 | 42.02 |
| 5 km | 0.31 | 23.35 | 29.55 | 37.00 | 41.46 | 50.90 | 51.20 |
| **Arkengarthdale** | | | | | | | |
| Lump | 0.00 | 32.27 | 43.35 | 55.16 | 59.67 | 73.18 | 76.78 |
| 10 km | 0.04 | 39.32 | 51.18 | 51.31 | 59.64 | 71.47 | 71.93 |
| 5 km | -4.28 | 39.25 | 51.48 | 61.22 | 65.81 | 82.59 | 75.84 |

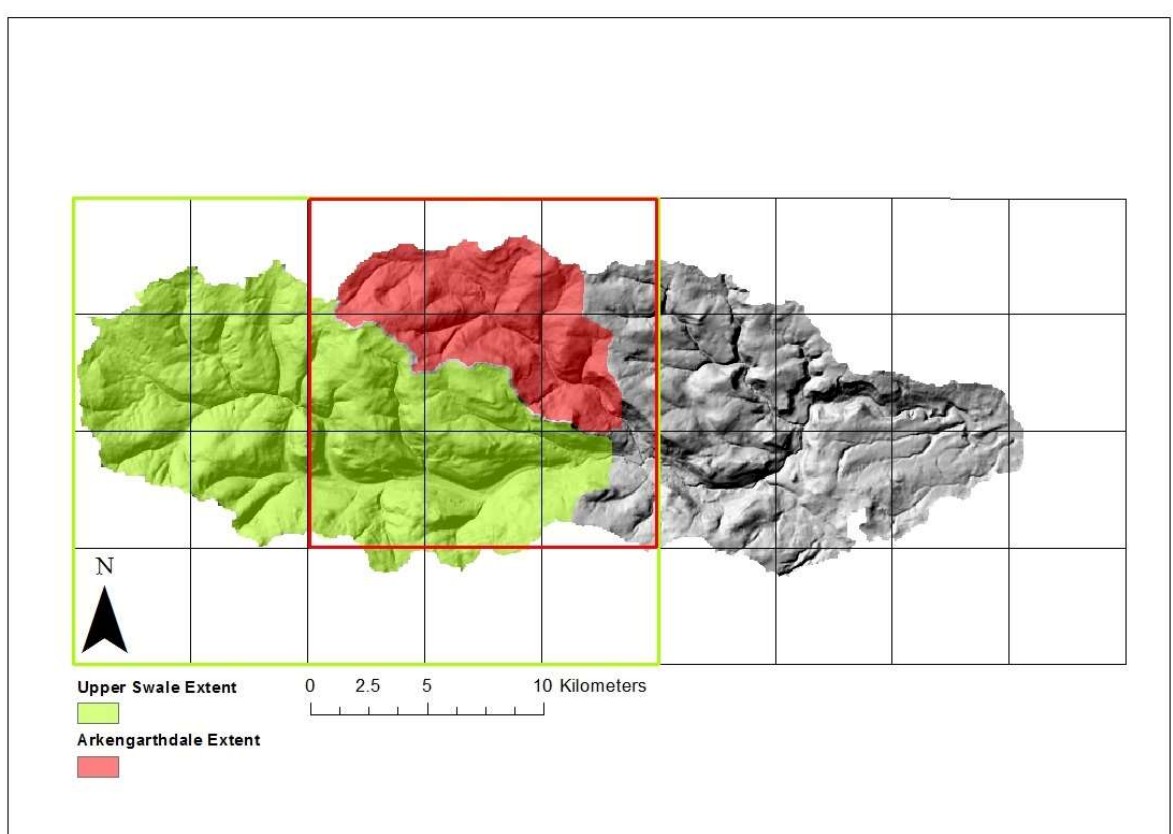

*Figure 1. Map showing the extents of the three test basins with the Upper Swale in green, and the Arkengarthdale extent in red. Additionally the 5 km rain radar grid cells overlaying the three basins are shown – coloured according to the basins they cover.*

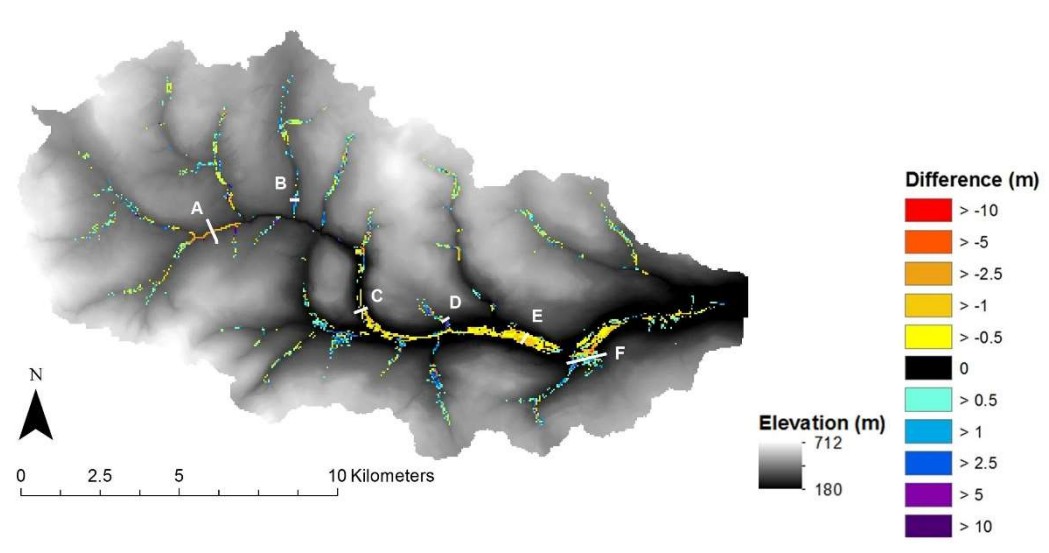

*Figure 2. DEM of Difference for the 1000 Year Swale Test. The differences shown are elevations from the 24hour - Lump simulation minus the elevations from the random 1 0.25 hour - 5km. Cross sections (Figure 3) are marked A-F. Yellows to reds indicate where the first (24 hour – Lump) simulation has eroded more/deposited less than the second (random 1) simulation. Blues indicate more deposition/less erosion.*

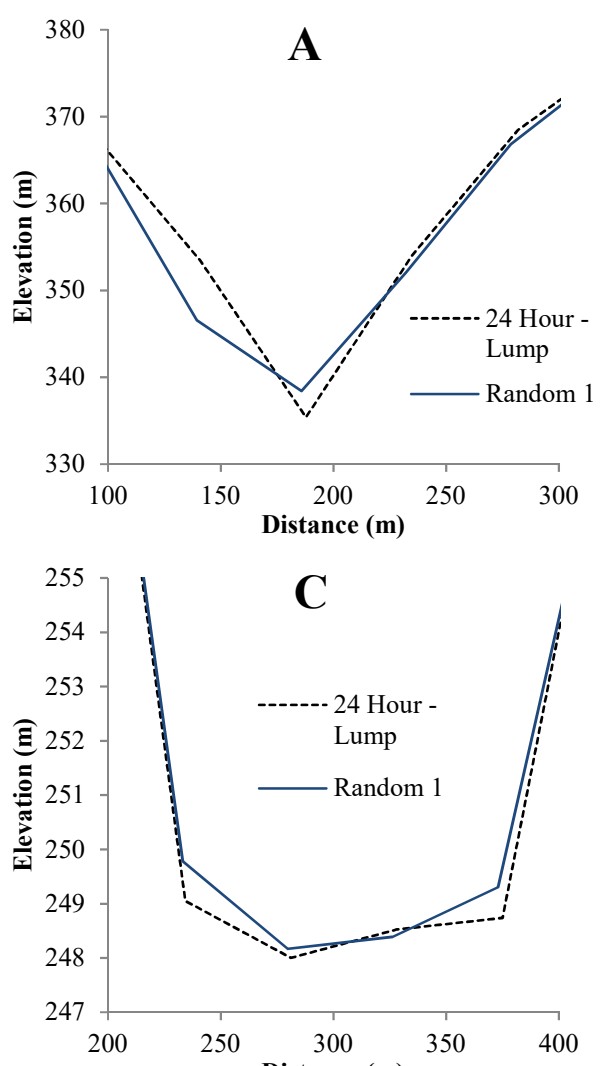
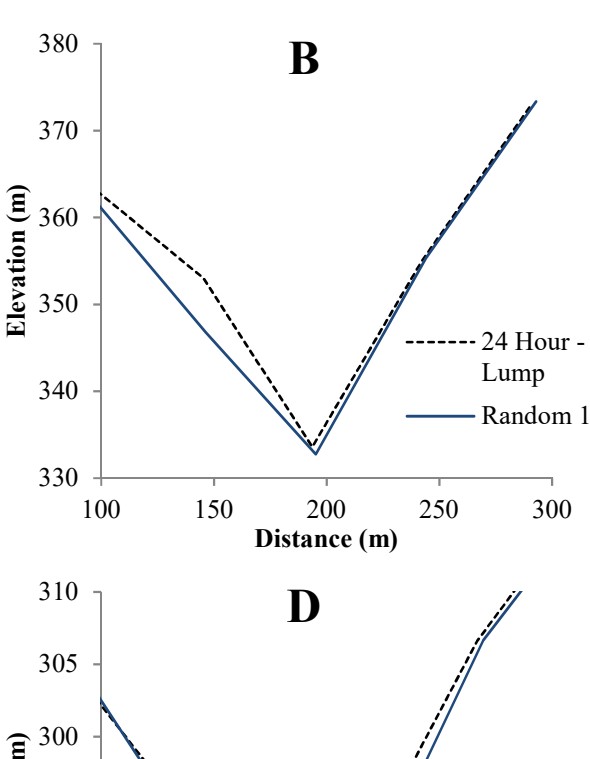
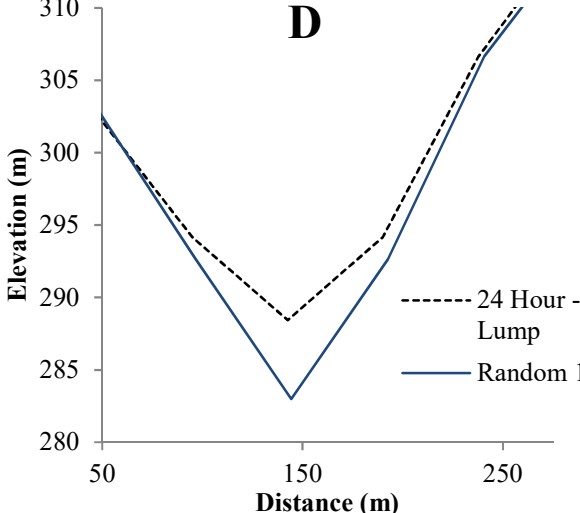

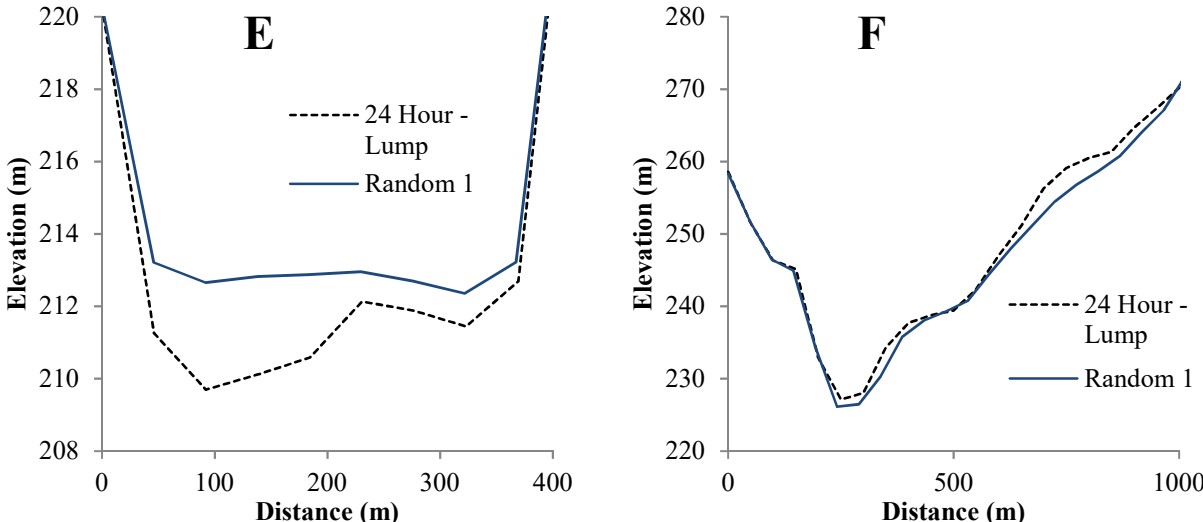

*Figure 3. Cross sections identified in Figure 2.*

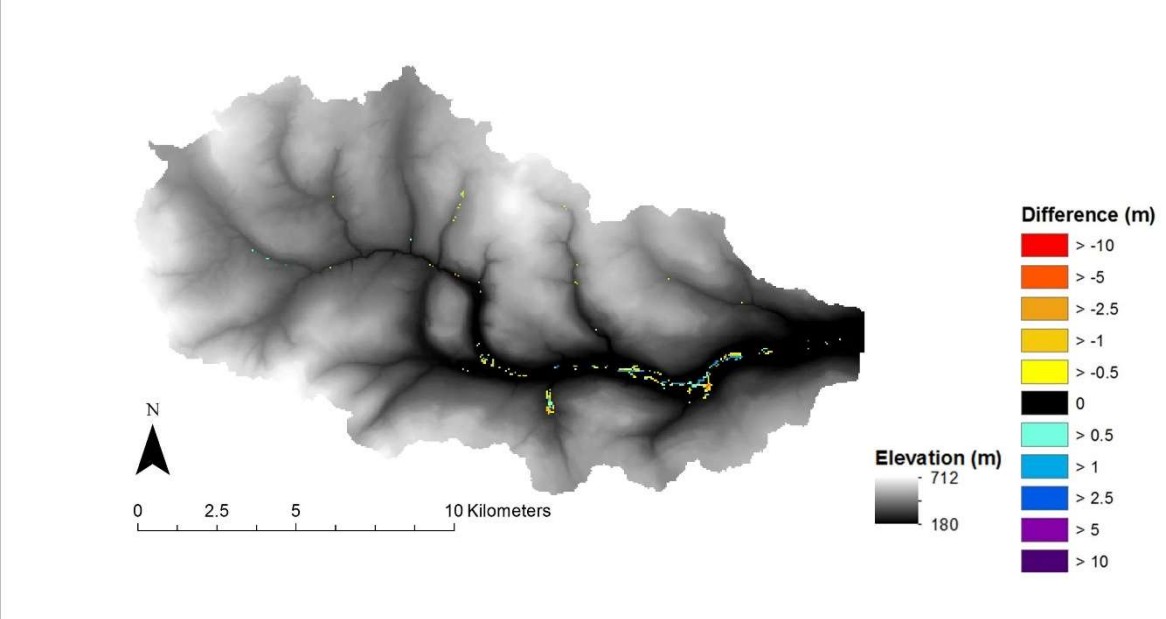

*Figure 4. DEM of difference between 1000 year random runs 1 and 2.*

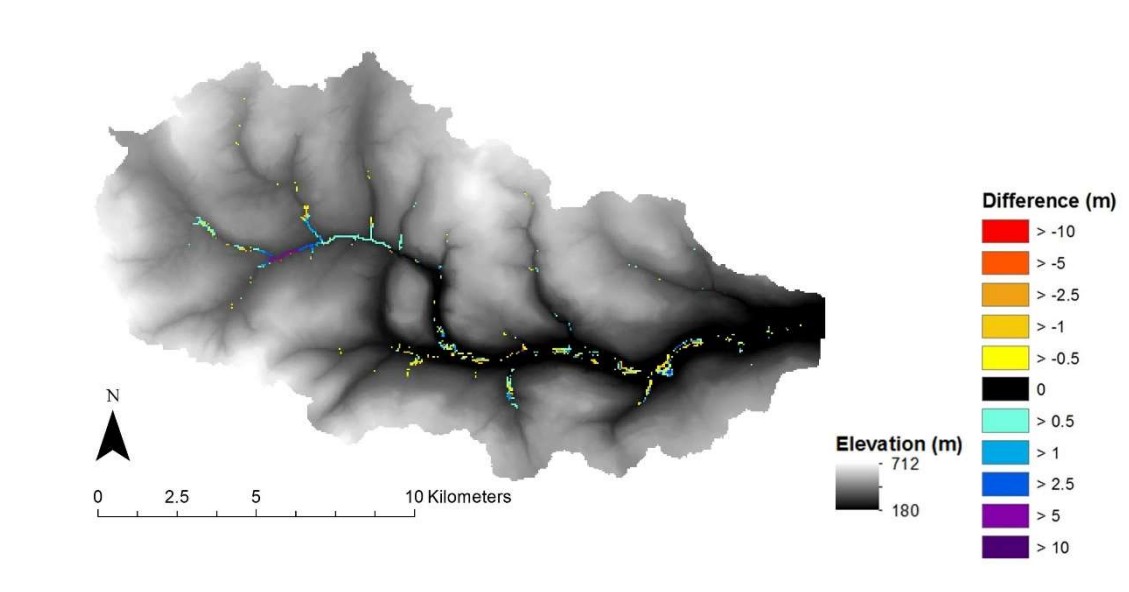

*Figure 5. DEM of Difference (DOD) between 1000 year random 1 and the 0.25 hour - 5km resolution simulation.*

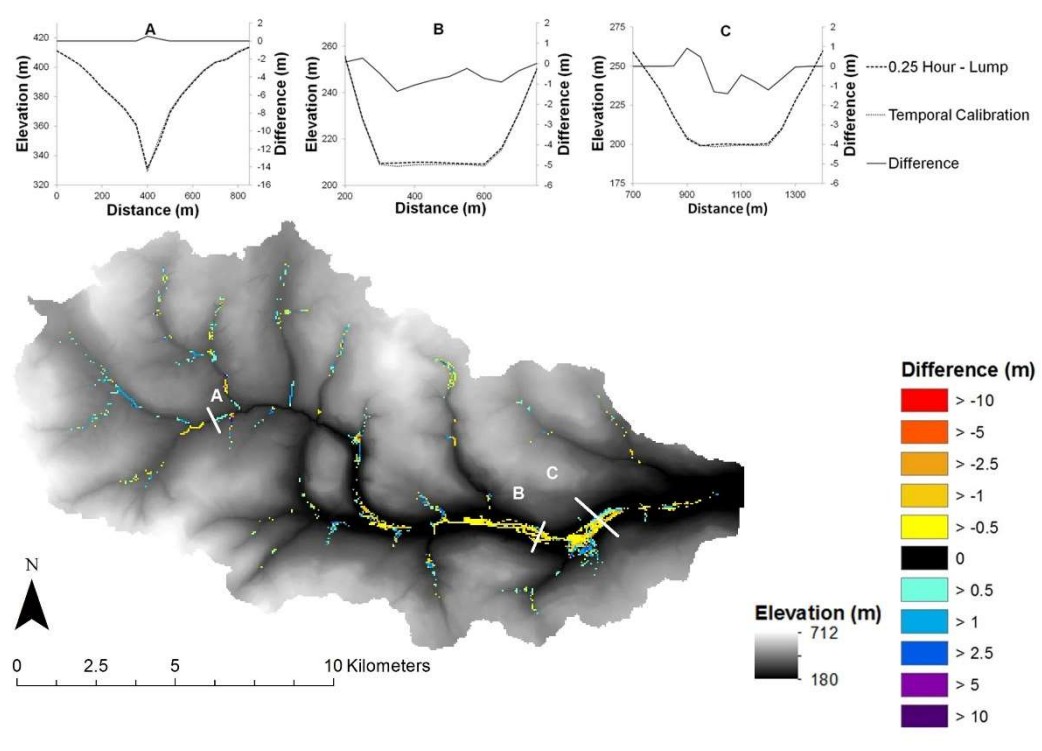

*Figure 6. DEM of Difference for the adjusted temporal resolution comparison. The differences shown are elevations from the 24 hour - Lump (2.0 factor) simulation minus the elevations from the 0.25 hour - Lump. Yellows to reds indicate where the first (24 hour – Lump 2.0) simulation has eroded more/deposited less than the second (0.25 hour - lump) simulation. Blues indicate more deposition/less erosion.*

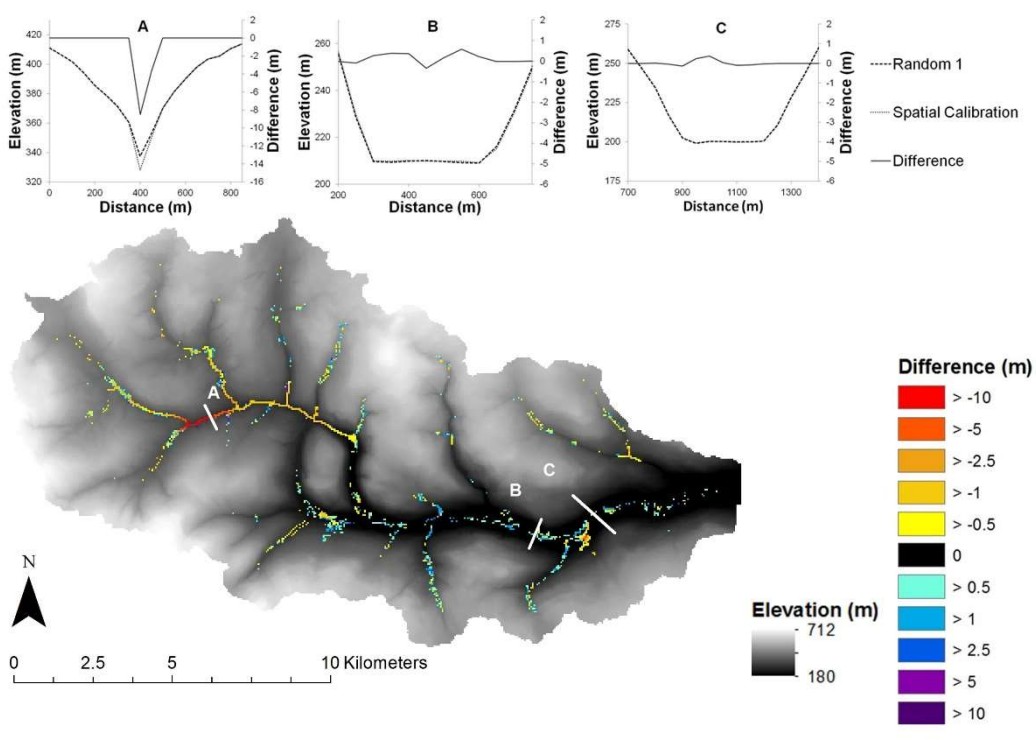

*Figure 7. DEM of Difference for the adjusted spatial resolution comparison. The differences shown are elevations from the 0.25 hour – Lump (1.1 factor) simulation minus the elevations from the 0.25 hour – 5 km. Yellows to reds indicate where the first (0.25 hour – Lump 1.1) simulation has eroded more/deposited less than the second (0.25 hour – 5 km) simulation. Blues indicate more deposition/less erosion.*

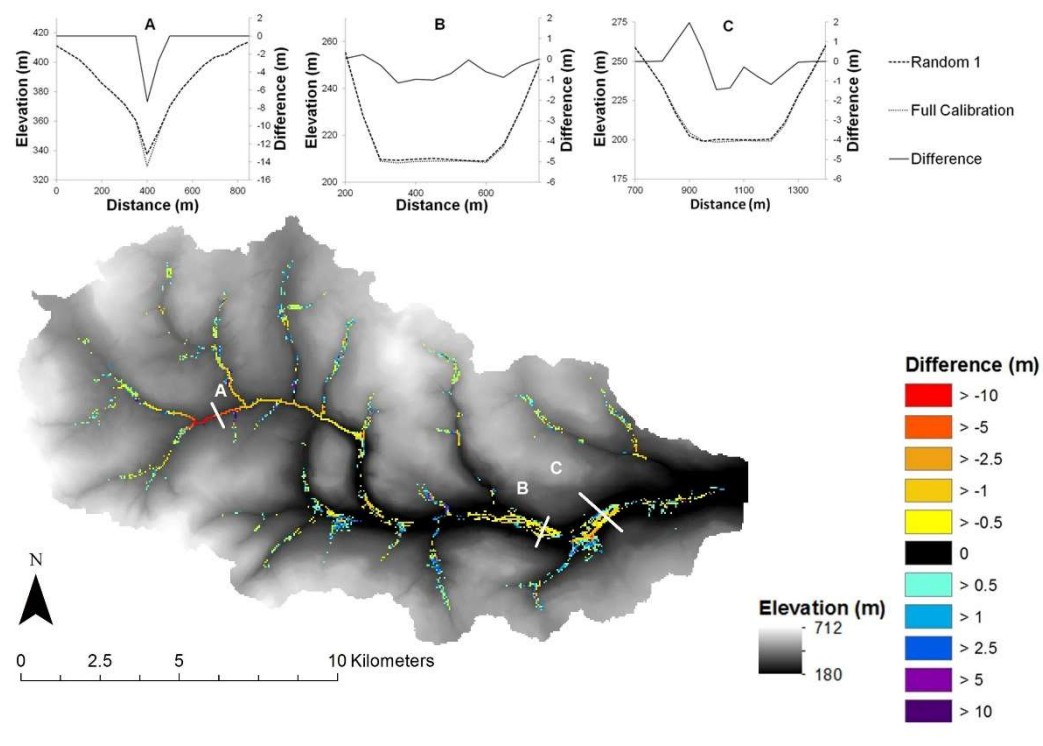

*Figure 8. DEM of Difference for the adjusted temporal and spatial resolution comparison. The differences shown are elevations from the 24 hour - Lump (2.2 factor) simulation minus the elevations from the 0.25 hour – 5 km. Yellows to reds indicate where the first (24 hour – Lump 2.2) simulation has eroded more/deposited less than the second (0.25 hour-5 km) simulation. Blues indicate more deposition/less erosion.*

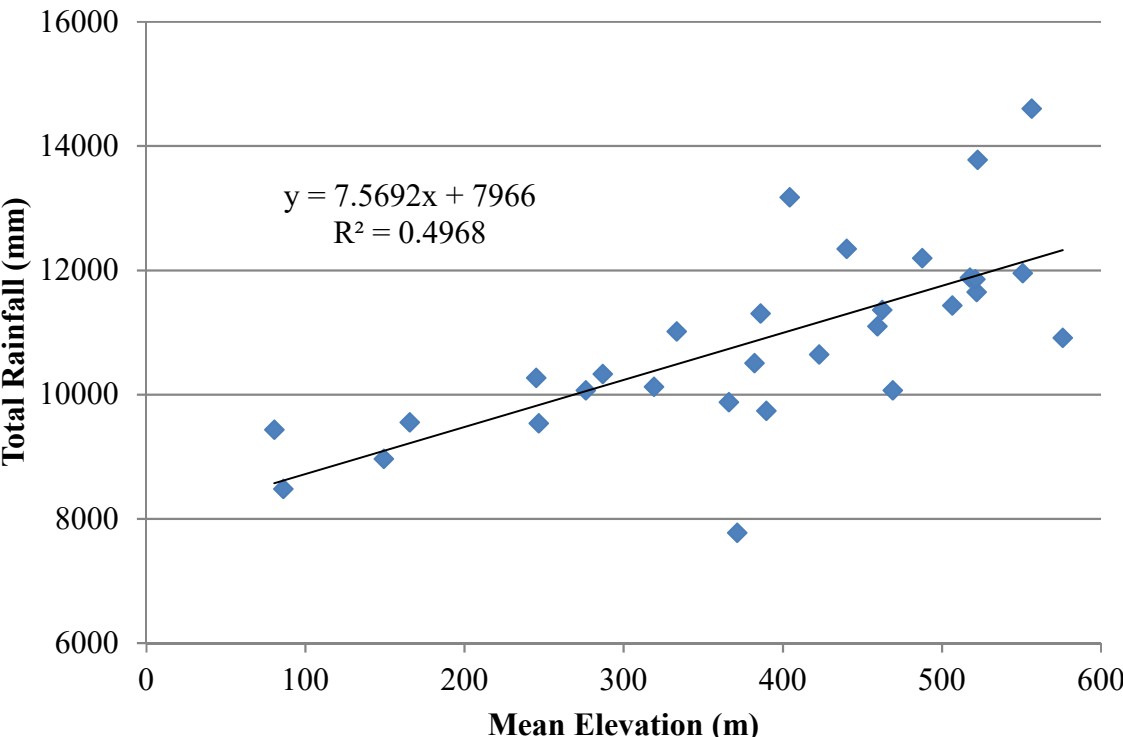

*Figure 9. Relationship between the total rainfall and mean elevation for each 5 km pixel within the Complete Swale basin.*

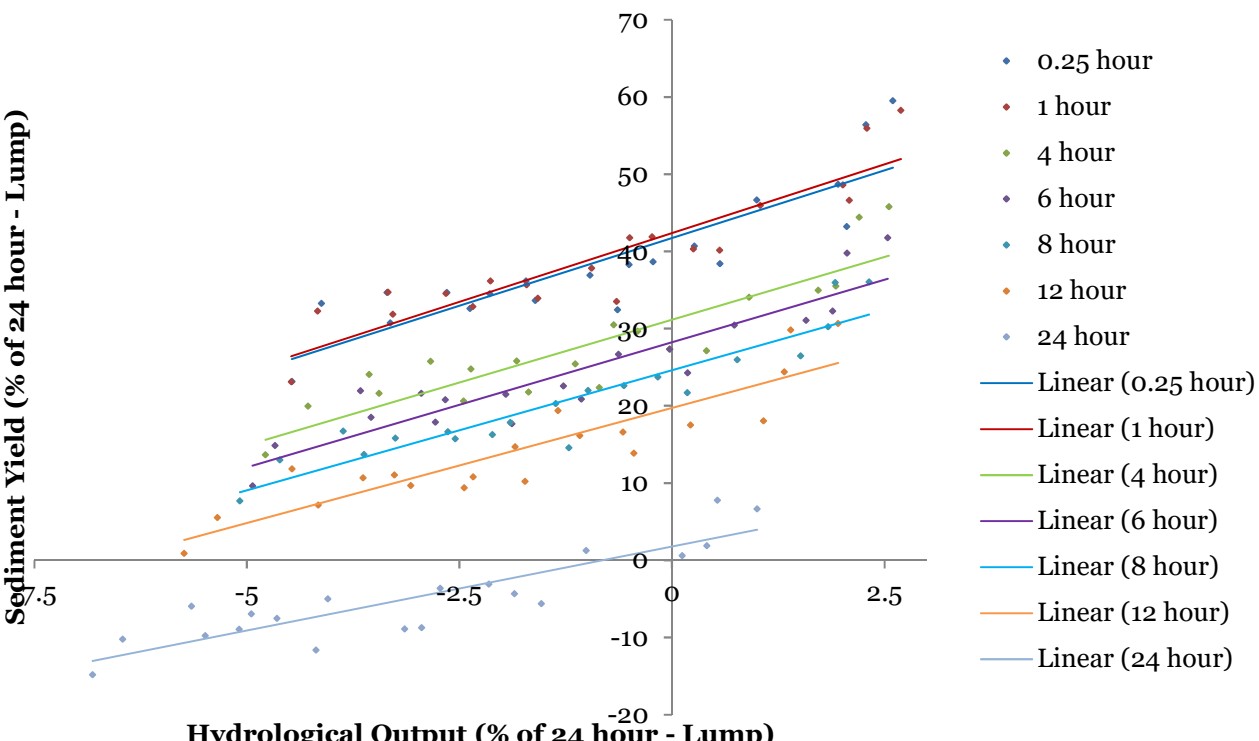

*Figure 10. The relationship between hydrological output and sediment yield from each temporal resolution, based on outputs of the 20 jumble ensembles of the Upper Swale catchment.*

