# Peer review of "The sensitivity of landscape evolution models to spatial and temporal rainfall resolution"

_Earth Surface Dynamics, 2016_

## Referee Comment (RC1) · Anonymous Referee #1 · 3 Mar 2016

General comments:

This paper deals with a very interesting and relevant question for the scientific community working on sediment transfers in mesoscale river basins: how do the spatial and temporal resolution of the meteorological forcing impact modelled sediment yields? While this issue has already been addressed from a purely hydrological standpoint, it remains understudied in modelling approach dealing with landscape evolution and soil erosion. However it seems to me that the conclusions raised by the authors are not supported by enough simulations. My main concern is about the potential effect of changes in soil hydrological properties (spatially and temporally) as the spatiotemporal resolution of rainfall is changed. This is not at all considered by the authors in

their simulations while they recognise at the end of the discussion that it may change considerably the sensitivity of landscape evolution models to rainfall resolution. As hydrological properties might be scale-dependant, changing only the spatiotemporal resolution of rainfall between runs without considering potential scale interactions between rainfall and soil behaviour may lead to erroneous conclusions on the sensitivity of landscape models. I know that adding runs in which the soil properties are randomly changed (m and K parameters) will need considerable additional computation time but the conclusions of the paper would be more supported and strengthened. Concerning the structure of the manuscript, the result section is very short and could be expanded, particularly if additional simulations are presented. The discussion section is rather heterogeneous in answering the 3 research questions written in the introduction. Section 4.2, addressing question 2, is very short and does not fully address it, as the authors recognise that more simulations would be required. Also Section 4.3, addressing question 3, is not supported by the data (no reference to them). I would suggest focusing the results and the discussion on question 1. If this question is fully addressed according to the above mentioned issues dealing with the hydrological basin properties, this would represent a substantial contribution to earth surface mass transfers. For those reasons, I do not recommend acceptance of the manuscript in its present form.

Specific remarks:

P2 L31-33: "Improved model performance" is not only "tempered by increased uncertainty surrounding precipitation data", but by the uncertainty in the budget of precipitation (P) versus infiltration (I) or storage in the soil.

P3 L8-9: I fully agree with the authors here. This also refers to the ability to simulate correctly P-I budgets, spatially and temporally. As argued in the general comments, the authors should try to address that issue in their numerical sensitivity analysis.

P4 L30: I found the description of the model spatial discretisation quite confusing (not sure to have completely understood yet), mentioning here "area lumped parameters"

and later (P5 L18) "grid cell size Dx" without giving any typical size for Dx. Is it the DEM resolution (50m) mentioned P7 L31? Could the authors try to be more specific on spatial discretisation and if possible limit the reference to previous papers to very specific model details that are not essential for that study?

P6 L1-3: As far as I understand, the hydrological model is adapted to the rainfall grid. Thus I agree with the authors that it enables having different levels of storage and runoff in each cell, but only due to rainfall variations. Varying also m and K would also create variations (i.e. P-I budget), but the authors kept constant those parameters (P8 L3).

P7 L29: How can the authors be confident in their conclusions with only 2 additional long term random simulations?

P8 L1 : Which initial grainsize distribution was used to run the 30 year model used as an initial condition? Which grain sizes are given in Table 3?

P8 L16: "considerable differences": this is not new and references should be given to situate these data.

P8 L21-23: I agree that these changes are minor. Could they be significantly different if m and K parameters were also randomly changed from one run to another?

P8 L28: "also drains an additional tributary". This may be critical. What is the drainage area of this tributary? How are the authors confident with that comparison between model and measurements?

P9 L11 : Very little difference is observed between random 1 and 2. This relates to a previous comment. Are 2 random simulations enough? Why not having done more, as presented later with the jumbled runs (P10 L9) for answering another question. Could these differences be more important if additional random simulations were performed? Otherwise this result (little difference) contradicts somehow with the results in Figure 7 showing a great dependence of the sediment yield to the rainfall allocation. Overall, it seems to me that the authors tried to address too many questions in the paper without

running enough simulations to address each of them.

P10 L8-9: if this issue is so important, it should be introduced as an objective of the paper. As written it appears like an additional side issue. The description of these jumbled runs should be added in the method section and removed from the discussion.

P10 L5-10: I fully agree with the authors that it is a major limitation to this study. Thus I recommend the authors to try to assess how m and Ks variations could impact the sensitivity analysis as it will help to generalize their findings.

P10 L10: Why 20 different records? Does this number has an impact on the range covered in Figure 7 (i.e. from -7 to 2,5% for X-axis and from -15 to 60% for Y-axis)?

P18 Table 3: Evaporation was set to 0. How does it impact the conclusions? K is missing in this table.

P27 Figure 6: Total rainfall : The authors should specify over how many years.

P28 Figure 7: Why was this analysis done on the upper Swale only? The complete basin is characterised by more rainfall cells and would have probably exhibited more variations in the random redistribution of rainfall (see author's comment in section 4.2, line 5). I find this figure very interesting in addition to the results from Table 8 for example, as it clearly shows the impact of the jumbled runs. This sensitivity of the sediment yield to different spatial and temporal distribution of the rainfall raises again the question : would this sensitivity be the same if also m and K parameters were included in a wider jumbled run numerical analysis.

Technical correction:

P2 L15 : delete reference at the end of the sentence P3 L19-20 : can not find those three references in the reference list P3 L28 : Coulthard et al. (2013a) P5 L4 : Add units for Qtot P6 L11-13 : n (Manning) is missing in the list P7 L29: were then compared P28 Figure 7: homogenize the colors for 6 hour (yellow and purple)

**ESurfD**

Interactive
comment

---

## Short Comment (SC1) · 3 Mar 2016

The authors present an interesting and detailed investigation into the sensitivity of landscape evolution models to the spatio-temporal resolution of rainfall data input. I don't wish to preempt the responses of the other reviewer(s), but I offer a few comments on the paper that the authors may wish to remark on or consider in their final manuscript.

**Data Source**

The authors have chosen to use rainfall radar data from the Met Office NIMROD system (Section 2.3, page 7, l.7), an appropriate choice for a study of this nature.  A minor point, but several different variations of this data prod-

uct are available, and it may help readers and future investigators to clarify precisely which data source was used – either the UK composite product, or the single site-specific radar data source from the nearest radar station to the Swale basin? (Both are available at 5km resolution). There are small processing differences between the two products (see http://browse.ceda.ac.uk/browse/badc/ukmo-nimrod/doc/radar_products_description.pdf). The citation could then be clarified appropriately, e.g.:

Met Office (2003): 5 km Resolution UK Composite Rainfall Data from the Met Office Nimrod System. NCAS British Atmospheric Data Centre.

As apposed to the current citation given which is a generic one for all the NIMROD rainfall radar datasets.

**Orography vs Spatial Resolution**

In section 4.1 (page 10, first paragraph) the authors discuss whether the increased erosion rates are due to the cumulative effects of orographic enhancement of rainfall, or purely due to the spatial resolution increase. They describe 'jumbling' the 5km rainfall data cells to produce a shuffled re-distribution of rainfall. However, is it possible that the jumbling of grid cells could have produced an pseudo-orographic effect in some of these jumbled simulations? In other words, a truly random shuffling of rainfall grid cells should be just as likely to produce some loose form of structure in the rainfall data as it is to produce a rainfall data set with a high degree of variability between neighbouring rainfall grid cells. Did the authors inspect the shuffled data to see if this had occurred? If so, this could be clarified, perhaps at an appropriate point the Methods section. Otherwise, I'm not sure that Figure 7 in particular supports the assertion that rainfall *spatial* resolution alone is responsible for increased sediment yields (and not due to any unintended rainfall pattern within the 'jumbled' data).

As a general comment regarding orographic rainfall effects (and perhaps as a corollary to the above comment), it is questionable whether this rainfall data, even at 5km

resolution, could sufficiently resolve smaller-scale orographic detail in rainfall in such a relatively small basin (Golding, 2000; Smith et al., 2015). Referring to Figure 1, only three rainfall grid cells (the highest rainfall resolution) cover the catchment in a North-South direction. In the East-West direction (c.30km), a general orographic gradient may well be resolved, however, due to sufficient grid cells in this dimension. Using a higher resolution rainfall radar product (Met Office, 2003) might be illuminating when investigating smaller basins (e.g. Valters et al., 2015).

**Unlimited Sediment Supply**

The authors use a model set-up with no bedrock layer (Section 2.3, page 7, l.31). I assume this means that there is effectively an unlimited supply of sediment during the model simulation. While it might be beyond the scope of this study to start considering transport-limited vs. detachment-limited scenarios, it could be useful to readers for the authors to discuss whether this could have an effect on their results. I.e. is it appropriate for the type of landscape in the Swale basin, particularly in its upland reaches (Howard, 1994), and would having limited sediment depth potentially have a limiting effect on sediment yields at higher rainfall resolutions? A brief comment on these aspects perhaps under 'Discussion' or 'Limitations' could clarify.

Thanks again to the authors for presenting an engaging study into the effects of rainfall resolution on landscape evolution models. I look forward to reading the final version of the manuscript in print.

**References**

Golding, B. W. (2000). Quantitative precipitation forecasting in the UK. Journal of Hydrology, 239(1), 286-305.

Howard, A. D. (1994). A detachment-limited model of drainage basin evolution. Water resources research, 30(7), 2261-2285.

Met Office (2003): 1 km Resolution UK Composite Rainfall Data from the Met Office

Nimrod System. NCAS British Atmospheric Data Centre,

Smith, S. A., Vosper, S. B., Field, P. R. (2015). Sensitivity of orographic precipitation enhancement to horizontal resolution in the operational Met Office Weather forecasts. Meteorological Applications, 22(1), 14-24.

Valters, D., Brocklehurst, S., Schultz, D. (2015). Sensitivity of hydro-geomorphic processes to catchment-scale variations in rainfall distribution. In EGU General Assembly Conference Abstracts (Vol. 17, p. 9909).

---

## Author Comment (AC1) · 3 Mar 2016

Thank you for the review – we hope we can use the online discussion to clarify one of the main points made before submitting a more formal response when all of the reviews are in.

Our query focuses around the comment "My main concern is about the potential effect of changes in soil hydrological properties (spatially and temporally) as the spatiotemporal resolution of rainfall is changed. This is not at all considered by the authors their simulations while they recognise at the end of the discussion that it may change considerably the sensitivity of landscape evolution models to rainfall resolution. As hydrological properties might be scale-dependent, changing only the spatiotemporal

resolution of rainfall between runs without considering potential scale interactions between rainfall and soil behaviour may lead to erroneous conclusions on the sensitivity of landscape models. I know that adding runs in which the soil properties are randomly changed (m and K parameters) will need considerable additional computation time but the conclusions of the paper would be more supported and strengthened"

As a simple description, a normal lumped application of Topmodel contains a store of water, release from which is controlled by the m parameter – and whether or not this is treated as runoff by k. In a regular – catchment wide lumped application, there is one store, one m value and one k value for everywhere. M and k are kept constant throughout operations of Topmodel – unless you are representing, for example, a change in land cover.

In our application, each 5km climate or hydrological cell contains a separate version of Topmodel – with its own store and (if required) m or k value (though in this study we deliberately keep these the same). Therefore the soil hydrological properties (the soil water store in Topmodel) of each cell are independent – so there is a spatio-temporal change in soil moisture across the basin (albeit in 5km cells). There are no sideways movements of water between adjacent climate/hydrological cells. Changing m or k for each cell in our application - would be akin to altering the land use within each cell which is not the focus of this paper. You could randomly change the m and k parameters randomly in each climate/hydrological cell – and repeat several (10's or even 100's) of times with the same rainfall patterns – to see whether the effect of rainfall resolution persisted through random variable changes. But would this really tell us any more than running it with the same m and k across the basin? If you randomly vary the m and k spatially and repeat enough times, then the average values of m and k will become constant and thus average water/sediment values will have the same output. Is this what is requested in the review?

There is of course a need to look at the sensitivity of models to both spatial and temporal changes in precipitation AND land use – but in this paper we have focused on

**ESurfD**
* * *
Interactive
comment

just one. This is to (a) make the experimental design simpler and (b) because spatial changes in land cover is really a different research/science question that we have answered in an additional paper (accepted - in press in another journal).

We hope this addresses the issue raised by the reviewer – but if we have misunderstood the above point then we would be very grateful if the reviewer could give some clarification?

Tom Coulthard, Chris Skinner.

---

## Short Comment (SC2) · 4 Mar 2016

Thank you for your comments, Declan. There are useful and are appreciated. I will attempt to address these here and in the final manuscript.

Data Source

You are entirely correct – we used the UK Composite data and will change references to this in the final manuscript.

Orography V Spatial Resolution

I will address the second part of your comment first. This is correct, this study looks at

a reasonably small catchment, yet in comparison the 5km rain grid cells are still coarse. However, as shown in Figure 6, there was an observed relationship of higher rainfall totals in grid cells with a higher mean elevation. One of the significant advantages of increasing the spatial resolution of the rainfall input is that you get a better representation of local variations, such as orographic effects. Increasing the spatial resolution further will only improve the representation of these variations further, yet also increase the uncertainty within the rainfall product.

The purpose of the 'jumbled' dataset was to investigate whether the same could be observed when the spatial distribution of rainfall was changed, disrupting any possible orographic effects. In these tests the spatial resolution was kept constant, but the rainfall intensities from each grid cell reassigned to another, and the temporal resolution varied. Figure 7 showed that in spite of changing patterns of rainfall here were clear step changes in sediment yields with each temporal resolution (except between 0.25 and 1 hour), and that these sediment yields were similar to that of the non-jumbled record – this indicates that it is the spatial resolution of the rainfall input which is influencing the sediment yield, and not a specific distribution of that rainfall. With this point, you are probably correct in that a pseudo-orography will exist, with some pixels still receiving a higher rate of rainfall than others, just not related to the elevation. The concluding remarks of the first paragraph of Page 10 should therefore be edited to say "Therefore, this strongly indicates that it is the spatial and temporal resolution and not any specific distributions of the data that are responsible for increased sediment yields previously described." Further similar edits will be made to this section accordingly.

Unlimited Sediment Supply

Yes, this experiment did assume an unlimited sediment supply. This is appropriate within the aims and objectives of the tests. This study is meant as an experiment looking at the sensitive of the model to the spatiotemporal resolution of the rainfall input, and data from the Swale were used to run this test as they had been used in the past by the authors and were well understood. However, it is not meant to be a case

study of the Swale basin or an attempt to accurately predict sediment yields from the catchment. This reflected in various aspects of the experiments, not just limited to a lack of bedrock representation, but also the lack of vegetation or land use variations too. All could act to either dampen or magnify the sensitivities observed here, and would make ideal foci for future studies (some already are). However, this could be stated more clearly throughout the manuscript and will be in the next iteration.

---

## Referee Comment (RC2) · Anonymous Referee #2 · 6 Mar 2016

Review of "The sensitivity of land scape evolution models to spatial and temporal rainfall resolution" Coulthard and Skinner

This paper examines the effect of temporal and spatial resolution has the erosion and landform evolution predictions of a LEM. The broad conclusions of the paper are a worthwhile contribution but the discussion misses some important points and previous work, and misrepresents previous work by other authors.

First looking at the question of spatial resolution. It's rather hard to judge the results without some idea of what the spatial pattern of rainfall is in the 10 year record and how persistent this pattern is over the 10 year period. A couple of thought experiments will clarify my concerns. 1. Imagine now that the pattern remains exactly the same

over the 10 year period (i.e. the amount of rainfall over the catchment changes from year to year, but the pattern of this rainfall is exactly the same form year to year). Then the random redistribution of rainfall in space will be completely invalid since what is required a random resampling of the rainfall in each year. This is an extreme case of orographic rainfall. 2. Imagine now that the pattern is completely uncorrelated from year to year and from 5km pixel to 5km pixel. In this case the random redistribution will be OK and any changes will simply result in random noise in the erosion and landform results. The authors have failed to justify that the differences they observe are anything other than random effects.

The way the question about time resolution is posed shows a misunderstanding of some of the solutions that other workers have used to address the problems highlighted of differences in mean erosion rate observed by the authors. There is no question that high time resolution runoff series results in significantly increased in erosion rates. The reviewer has also seen this in his our erosion computations and the 100% increase from daily to 0.25 hour accords with our own, unpublished, experience. This is because of the nonlinear dependence of the erosion time series on the runoff time series. A simple first order second moment analysis of the erosion time series shows this. Consider an erosion equation that is dependent on the square of discharge (approximately the dependency of Einstein-Brown sediment transport equation) $E=bQ^2$ (1)

If Q is now a random value with mean $Q^*$ and variance SQ. A second order first moment analysis of this equation yields

$E=b(Q^{*2}+SQ)$ (2)

So that the erosion is higher than that where there is no variation in Q by a factor

$(1+SQ/Q^{*2})$ (3)

This analysis shows that the erosion rate when you allow for randomness versus where

you average out the variability will always be higher and the percentage increase is a function of the coefficient if variation of the runoff series. My own observation is that this factor can easily by a factor 2 going from a daily runoff series to a 15 minute runoff series for a small catchment (i.e. the erosion will increase by 100%). The appearance of variance in equation (2) comes solely from the square dependence in equation (1). If equation (1) were a power of 1 (i.e. linear) then the variance term does not appear and the sub-daily variability would have no impact on the mean erosion rate.

Finally, the authors quote Hancock papers (2000,2002,2010) as examples where the long time resolution of the timesteps in the landform evolution model will yield significant underestimates of the erosion. This assertion is categorically incorrect and reflects a lack of understanding of how the model parameters were developed for these papers. I'm surprised at this because the first author has been collaborating for some time with Hancock. The parameters used in the Hancock papers are based on a calibration procedure described in Willgoose and Riley (1993,1998)

Willgoose, G. R., and S. J. Riley (1993), Application of a catchment evolution model to the prediction of long-term erosion on the spoil heap at Ranger Uranium MineRep. Open File Report 107, The Office of the Supervising Scientist, Jabiru.

Willgoose, G. R., and S. J. Riley (1998), An assessment of the long-term erosional stability of a proposed mine rehabilitation, Earth Surface Processes and Landforms, 23, 237-259.

In brief this process was 1. A conceptual rainfall-runoff model (with much the same capability as LISFLOOD) was calibrated to rainfall-runoff-erosion plot studies at the time and space resolution of the data (minutes and 100 sq metres) 2. A multiple regression was developed between sediment load, discharge and slope from the plot studies. 3. The rainfall-runoff model was then scaled up to the landform using a low resolution DEM of the site (about 1000 nodes) and 30 years of pluviograph data at 15 minute resolution was used to generate a 30 year runoff time series. 4. This 15 minute resolution

time series was then used to generate a 15 minute sediment transport series using the regression. 5. This 15 minute erosion series was then lumped up to the annual level and "effective" parameters where developed that gave the same average and area and slope dependence at the yearly time step as the 15 minutes erosion time series. These are the parameters that are used in the annual time steps.

Now there is no doubt this was an extremely compute intensive task. In 1992 when this work was done it took about 4 weeks of CPU time on a high end workstation to generate the time series in step 3. This calibration has been used as the basis for other sites studied by Hancock.

The key difference between what was done by Willgoose and Riley (1998) (hereafter W&R) and in this paper is that the authors have explicitly included the randomness of the hydrology timeseries within the LEM, while in W&R this has been averaged out in the derivation of the effective parameters.

Finally on bottom of p10 and top of p11 the author contemplates whether there is a "compensatory factor or exponent". Indeed this is what the "effective parameters" in the approach of W&R do.

So in conclusion if we go to the plots of changes when using different averaging periods, the lower erosion rate observed by the author for low resolution rainfall is to be expected. But this can be adjusted by the use of "effective parameters" as done in W&R. The more interesting question, but unfortunately not addressed by the authors, is if the average erosion rate for all the different time resolutions were adjusted to give the same annual erosion are the landforms generated significantly different (i.e. does the higher rainfall resolution and explicit modelling of runoff events lead to fundamental differences beyond a general change in the calculated mean erosion rate).

---

## Editor Comment (EC1) · G. Govers (Editor) · 28 Apr 2016

Dear authors,

My apologies that it took a while to post this which is due to a misunderstanding. I think you have received two insightful reviews of the manuscript that you submitted. In my view the paper can be published in Esurf if these remarks (and the ones suggested by Declan Valters) are accounted for in a revised version.

The main comment of ref. 1 relates to the potential interaction of soil properties and resolution. I most certainly agree with the fact that this is a valid point that should be dealt with in a revised manuscript. However, it is in my view not necessary to carry out

a large number of additional simulations to investigate this as this would be a different research topic altogether. I do think though that you should discuss this issue (and its potential implications) in a revised version of the MS. You already addressed the issue along those lines in an author comment that can form the basis for the rewriting of this part of the MS

Reviewer #2 has concerns witht the way you deal with increased temporal rainfall resolution. As eroison is indeed a non-linear function of discharge/rainfall intensity non-linear effects are indeed to be expected. Reading your MS I do feel these remarks are important but may partially be caused by a misunderstanding of the procedure you used and of the aims of your study, which focuses on spatial patterns rather than total erosion amounts. Please clarify this in a revised manuscript: I am sure that this will also be of great help to other researchers who want to better understand your research.

Kind regards,

Gerard Govers, Associate Editor

---

## Author Comment (AC2) · 17 Jun 2016

**The sensitivity of landscape evolution models to spatial and temporal rainfall resolution:**

**Reviewer 2 Comments**

*We have made considerable changes to the MS based on this review – both including a number of additional references and in a series of new simulations to address the final point made by the reviewer. These are detailed below and in the revised MS.*

This paper examines the effect of temporal and spatial resolution has the erosion and landform evolution predictions of a LEM. The broad conclusions of the paper are a worthwhile contribution but the discussion misses some important points and previous work, and misrepresents previous work by other authors.

*We would like to thank the reviewer for their comments and to apologise for completely missing developments made by previous authors. Some of the comments by the reviewer refer to unpublished research examining the role of temporal rainfall resolution – and make complete sense as does the thought experiment outlined in the review. However, it is difficult to reference unpublished findings, but we have looked in some detail at the SIBERIA literature, finding a relevant section in a user manual and used this accordingly in the revised MS. The calibration process outlined in the Willgoose and Riley (1998) paper makes no direct reference to rainfall resolution – but having read the reviewers comments – and re-read the paper sections it is clear that this is part of the calibration process. We have added sections and re-worked parts of the paper to clearly acknowledge this. Our findings (with regard to temporal rainfall precipitation) certainly agree with those mentioned above – and this is duly noted.*

*We considered removing the temporal component of the model comparison and focusing on the spatial in the revised MS, but thought that our experiments still contained an important contribution as it looked at how the relationship changed through different resolutions as well as over different basins. Additionally it also allowed the combination of spatial and temporal rainfall resolution to be examined. Therefore, it represents a systematic investigation into rainfall spatial and temporal resolution.*

First looking at the question of spatial resolution. It's rather hard to judge the results without some idea of what the spatial pattern of rainfall is in the 10 year record and how persistent this pattern is over the 10 year period. A couple of thought experiments will clarify my concerns.

1. Imagine now that the pattern remains exactly the same over the 10 year period (i.e. the amount of rainfall over the catchment changes from year to year, but the pattern of this rainfall is exactly the same form year to year). Then the random redistribution of rainfall in space will be completely invalid since what is required a random resampling of the rainfall in each year. This is an extreme case of orographic rainfall.

2. Imagine now that the pattern is completely uncorrelated from year to year and from 5km pixel to 5km pixel. In this case the random redistribution will be OK and any changes will simply result in random noise in the erosion and landform results. The authors have failed to justify that the differences they observe are anything other than random effects.

*We have to be careful to consider that in reality rainfall is not random. It does have patterns (spatial and temporal) – and some of these temporal patterns should be retained otherwise the resampled/ chopped record is meaningless. Therefore, we have not randomly re-sampled during the year – as the rainfall is made of 'events' – here largely associated with frontal rainfall. It would be unrealistic to distintangle these events – so you would (for example) have one pixel of heavy rain pop up in the middle of a dry spell. We have done this to a degree by spatially 'mixing' every 10 years – but the mixed pixels are still in temporal sync with each other. This could be broken down into annual mixing – but over a 1000 year simulation would that really give a different solution from our one? As figure 4 shows, there is relatively little difference between two of our random 1000 year simulations.*

*What we have done by spatially mixing the rain cells every 10 years in a 1000 year run, is to show the aggregate of 100 mixed up, 10 year simulations (its an easier way than showing an average if you like). By having different mixed up runs that give very similar results spatially and in bulk yields (Figure 4) – yet clearly different from the non mixed up results (figure 5) - we show that we can remove any spatial bias in the patterns of rainfall we are using in these 1000 year simulations. This means that we can compare 5km spatially distributed (randomly mixed spatially every 10 years) to lumped rainfall simulations over the same period.*

*A neater solution to this issue would be to use a synthetic rainfall generator that also simulates spatial patterns of rainfall. These exist, though are relatively new and less tested than non spatial rainfall generation methods (e.g.* Peleg & Morin 2014*). Here, this would significantly expand the work required, scope and aims of the paper (in effect, it is another paper).*

*Peleg, N. & Morin, E., 2014. Stochastic convective rain-field simulation using a high-resolution synoptically conditioned weather generator (HiReS-WG). Water Resources Research, 50(3), pp.2124–2139. Available at: http://doi.wiley.com/10.1002/2013WR014836 [Accessed June 17, 2016].*

The way the question about time resolution is posed shows a misunderstanding of some of the solutions that other workers have used to address the problems highlighted of differences in mean erosion rate observed by the authors. There is no question that high time resolution runoff series results in significantly increased in erosion rates. The reviewer has also seen this in his our erosion computations and the 100% increase from daily to 0.25 hour accords with our own, unpublished, experience. This is because of the nonlinear dependence of the erosion time series on the runoff time series. A simple first order second moment analysis of the erosion time series shows this.

Consider an erosion equation that is dependent on the square of discharge (approximately the dependency of Einstein-Brown sediment transport equation)

$E=bQ^2$ (1)

If Q is now a random value with mean Q* and variance SQ. A second order first moment analysis of this equation yields

E=b(Q*ˆ2+SQ) (2)

So that the erosion is higher than that where there is no variation in Q by a factor

(1+SQ/Q*ˆ2) (3)

This analysis shows that the erosion rate when you allow for randomness versus where you average out the variability will always be higher and the percentage increase is a function of the coefficient if variation of the runoff series.

My own observation is that this factor can easily by a factor 2 going from a daily runoff series to a 15 minute runoff series for a small catchment (i.e. the erosion will increase by 100%). The appearance of variance in equation (2) comes solely from the square dependence in equation (1). If equation (1) were a power of 1 (i.e. linear) then the variance term does not appear and the sub-daily variability would have no impact on the mean erosion rate.

*This is a really interesting way of breaking down the issue for temporal rainfall – in our representation, erosion (with the addition of various parameters) is roughly the square of the velocity – so a similar relationship. We would have liked to include a similar breakdown in the revised MS – but would not want to make this look like our thoughts (and we cannot readily cite reviews). Hopefully, the quote from the SIBERIA manual we have included covers part of this (certainly the last para above).*

Finally, the authors quote Hancock papers (2000,2002,2010) as examples where the long time resolution of the timesteps in the landform evolution model will yield significant underestimates of the erosion. This assertion is categorically incorrect and reflects a lack of understanding of how the model parameters were developed for these papers. I'm surprised at this because the first author has been collaborating for some time with Hancock. The parameters used in the Hancock papers are based on a calibration procedure described in Willgoose and Riley (1993,1998) Willgoose, G. R., and S. J. Riley (1993),  Application of a catchment evolution model to the prediction of long-term erosion on the spoil heap at Ranger Uranium MineRep. Open File Report 107, The Office of the Supervising Scientist, Jabiru. Willgoose, G. R., and S. J. Riley (1998), An assessment of the long-term erosional stability of a proposed mine rehabilitation, Earth Surface Processes and Landforms, 23, 237-259.

In brief this process was 1. A conceptual rainfall-runoff model (with much the same capability as LISFLOOD) was calibrated to rainfall-runoff-erosion plot studies at the time and space resolution of the data (minutes and 100 sq metres) 2. A multiple regression was developed between sediment load, discharge and slope from the plot studies. 3. The rainfall-runoff model was then scaled up to the landform using a low resolution DEM of the site (about 1000 nodes) and 30 years of pluviograph data at 15 minute resolution was used to generate a 30 year runoff time series. 4. This 15 minute resolution  time series was then used to generate a 15 minute sediment transport series using the regression. 5. This 15 minute erosion series was then lumped up to the annual level and "effective" parameters where developed that gave the same average and area and slope dependence at the yearly time step as the 15 minutes erosion time series. These are the parameters that are used in the annual time steps.

Now there is no doubt this was an extremely compute intensive task. In 1992 when this work was done it took about 4 weeks of CPU time on a high end workstation to generate the time series in step 3. This calibration has been used as the basis for other sites studied by Hancock.

The key difference between what was done by Willgoose and Riley (1998) (hereafter W&R) and in this paper is that the authors have explicitly included the randomness of the hydrology timeseries within the LEM, while in W&R this has been averaged out in the derivation of the effective parameters.

Finally on bottom of p10 and top of p11 the author contemplates whether there is a "compensatory factor or exponent". Indeed this is what the "effective parameters" in the approach of W&R do.

So in conclusion if we go to the plots of changes when using different averaging periods, the lower erosion rate observed by the author for low resolution rainfall is to be expected. But this can be adjusted by the use of "effective parameters" as done in W&R.

The more interesting question, but unfortunately not addressed by the authors, is if the average erosion rate for all the different time resolutions were adjusted to give the same annual erosion are the landforms generated significantly different (i.e. does the higher rainfall resolution and explicit modelling of runoff events lead to fundamental differences beyond a general change in the calculated mean erosion rate).

This *is* a really interesting question – and we are grateful for the reviewer for suggesting this. In the revised MS we have now done just this – to adjust model runs (via a compensation factor in the sediment transport law) so very similar sediment yields (erosion rates) are generated over 1000 year simulations. Rather than try and tune all our simulations to the same erosion rate (and therefore to reduce the number of simulations needed) we adjusted some simulations (e.g. 15 min lumped) too match existing results (e.g. 24 hour lumped). This required an additional 30 simulations – each taking 4-8 weeks. This generated some really interesting findings – and as the reviewer suggested – does lead to considerable differences in the spatial patterns of erosion and deposition found within the basin.

These simulations and research, have resulted (in the paper) in additional sections in the methods, results, discussion and conclusions – and we think they significantly enhance the paper and its findings.

---

## Author Comment (AC3) · 17 Jun 2016

**The sensitivity of landscape evolution models to spatial and temporal rainfall resolution:**

**Reviewer 1 Comments**

This paper deals with a very interesting and relevant question for the scientific community working on sediment transfers in mesoscale river basins: how do the spatial and temporal resolution of the meteorological forcing impact modelled sediment yields? While this issue has already been addressed from a purely hydrological standpoint, it remains understudied in modelling approach dealing with landscape evolution and soil erosion. However it seems to me that the conclusions raised by the authors are not supported by enough simulations. My main concern is about the potential effect of changes in soil hydrological properties (spatially and temporally) as the spatiotemporal resolution of rainfall is changed. This is not at all considered by the authors in their simulations while they recognise at the end of the discussion that it may change considerably the sensitivity of landscape evolution models to rainfall resolution. As hydrological properties might be scale-dependant, changing only the spatiotemporal resolution of rainfall between runs without considering potential scale interactions between rainfall and soil behaviour may lead to erroneous conclusions on the sensitivity of landscape models. I know that adding runs in which the soil properties are randomly changed (m and K parameters) will need considerable additional computation time but the conclusions of the paper would be more supported and strengthened.

*We would like to thank the reviewer for their comments and thorough review. Aside from typo's and other minor points/clarifications, the main point the reviewer asks us to address is the interaction with soil properties and the balance between precipitation (P) and infiltration (I).*

*We agree completely with the reviewer that soil and land use properties might influence our results. However, the focus of this study is to examine just the impact of spatial and temporal rainfall resolution. In our parameterisation, hydrological factors that will change spatially are deliberately treated globally so we can look solely at the role of rainfall resolution. The experimental set up (e.g. having different hydrological areas defined by the rainfall grid resolution) is contingent upon the deliberately limited research questions we are asking – and to look at both soil properties and rainfall resolution would, we suggest, require a completely different model set up.*

*We believe that it is important to consider that basin hydrology – both in terms of soil properties – and the driving precipitation – is often dealt with incredibly simplistically in LEM's, if at all! Therefore, our motivation is to explore not just the sensitivity to resolution – but to show the difference between having no representation and some representation of a distributed hydrology in LEM's.*

*Since this paper was first submitted – we have also submitted (and now published in early view) an article that takes a tightly constrained look at the impact of spatial changes in the TOPMODEL m value on the geomorphic outputs over longer time scales* (Coulthard & Van De Wiel, 2016). *There is certainly a place for a study looking at both together. This could be looking at a combination of the two approaches opens up the CAESAR-Lisflood model to a framework of modelling using Hydrological Response Units (HRU), a common approach in semi-distributed hydrological models, such as Dynamic-TOPMODEL and SWAT. This allows rainfall, land cover and soil properties to be represented at higher resolution than a global lumped estimate, but divided into broadly hydrologically homogenous regions.*

*Whilst we have not carried out any additional research to answer the points raised by the reviewer above and below, we certainly accept their validity – and have added a section to the discussion/limitations section*

*Coulthard, T. J., & Van De Wiel, M. J. (2016). Modelling long term basin scale sediment connectivity, driven by spatial land use changes. Geomorphology. http://doi.org/10.1016/j.geomorph.2016.05.027*

Concerning the structure of the manuscript, the result section is very short and could be expanded, particularly if additional simulations are presented. The discussion section is rather heterogeneous in answering the 3 research questions written in the introduction. Section 4.2, addressing question 2, is very short and does not fully address it, as the authors recognise that more simulations would be required. Also Section 4.3, addressing question 3, is not supported by the data (no reference to them). I would suggest focusing the results and the discussion on question 1.

We have changed the research questions – and dropped number 2.

If this question is fully addressed according to the above mentioned issues dealing with the hydrological basin properties, this would represent a substantial contribution to earth surface mass transfers. For those reasons, I do not recommend acceptance of the manuscript in its present form.

Specific remarks:

P2 L31-33: "Improved model performance" is not only "tempered by increased uncertainty surrounding precipitation data", but by the uncertainty in the budget of precipitation (P) versus infiltration (I) or storage in the soil.

P3 L8-9: I fully agree with the authors here. This also refers to the ability to simulate correctly P-I budgets, spatially and temporally. As argued in the general comments, the authors should try to address that issue in their numerical sensitivity analysis.

See first comment. There is of course a need to look at the sensitivity of models to both spatial and temporal changes in precipitation AND land use – but in this paper we have focused on just one. This is to (a) make the experimental design simpler and (b) because spatial changes in land cover is really a different research/science question that we have answered in a subsequent paper.

P4 L30: I found the description of the model spatial discretisation quite confusing (not sure to have completely understood yet), mentioning here "area lumped parameters" and later (P5 L18) "grid cell size Dx" without giving any typical size for Dx. Is it the DEM resolution (50m) mentioned

Changed to say 50m.

P7 L31? Could the authors try to be more specific on spatial discretisation and if possible limit the reference to previous papers to very specific model details that are not essential for that study?

There are no references in this paragraph, the three before or the two after - think this relates to the overall model description.

P6 L1-3: As far as I understand, the hydrological model is adapted to the rainfall grid. Thus I agree with the authors that it enables having different levels of storage and runoff in each cell, but only

due to rainfall variations. Varying also m and K would also create variations (i.e. P-I budget), but the authors kept constant those parameters (P8 L3).

This is true - but the cells do not cover hydrologically homogenous areas and values of m and k would be difficult to determine. m and k are not scale dependent, therefore the use of global values here is justified, yet we acknowledge that adjusting these values to local conditions will change the outputs - this is beyond the scope of this study (which is motivated only by rainfall resolution).

P7 L29: How can the authors be confident in their conclusions with only 2 additional long term random simulations?

In each random scenario the rainfall distribution was reshuffled every ten years, producing a significant element of randomness to the full 1000 year record simulated. However, both these two random runs produced similar results to each other. Additional random scenarios could be run, yet this is computationally expensive and would yield similar results. The random runs were motivated by wanting to disrupt the spatial pattern in the ten year record, which is repeated 100 times in the long term runs - in reality, we could have been confident with just one test as this achieved this, but the only very minor variations between the two reinforce this further.

We can see that this may not be clear from the text – so this has been clarified in the methods section. Part of this is trying to explain how methods evolved in response to results – but within the methods section!

P8 L1 : Which initial grainsize distribution was used to run the 30 year model used as an initial condition? Which grain sizes are given in Table 3?

The grainsizes in Table 3 are those used to initialise the model with a global distribution, which was then spun up using a thirty year simulation. The spun-up grain size distribution was used for the tests. Text has been added to make this much clearer.

P8 L16: "considerable differences": this is not new and references should be given to situate these data.

Yes - this well established and long been understood. Line 16 has been altered in the manuscript to make this clearer.

P8 L21-23: I agree that these changes are minor. Could they be significantly different if m and K parameters were also randomly changed from one run to another?

Possibly, but as previously mentioned that is not the aim of this study.

 P8 L28: "also drains an additional tributary". This may be critical. What is the drainage area of this tributary? How are the authors confident with that comparison between model and measurements?

The purpose of the hydrology tests was not to assess the model's performance and skill at estimating the hydrological conditions, but to demonstrate the effects on the hydrology when driving the model with different resolutions of input data. We believe that the data presented do this effectively.

[Figure]

The image above shows the full extent of the Swale catchment we have used, plus the location of the Catterick Bridge gauge station and the missing drainage network. The system is predominantly Gilling Beck , which becomes Skeeby Beck before entering the River Swale. The information for the Catterick Bridge Gauge station (http://nrfa.ceh.ac.uk/data/station/info/27090) states it has a draingage area of 499.4km$^2$ - of which, the Complete Swale DEM is 415km$^2$ - so the missing area covers 17% of the drainage area. It's overall contribution to the discharge of the River Swale is not known.

P9 L11 : Very little difference is observed between random 1 and 2. This relates to a previous comment. Are 2 random simulations enough? Why not having done more, as presented later with the jumbled runs (P10 L9) for answering another question. Could these differences be more important if additional random simulations were performed? Otherwise this result (little difference) contradicts somehow with the results in Figure 7 showing a great dependence of the sediment yield to the rainfall allocation. Overall, it seems to me that the authors tried to address too many questions in the paper without running enough simulations to address each of them.

The jumble runs involve the shuffling of the rainfall distribution just the once. Random 1 and 2 each involve 100 reshuffles of the rainfall distribution and therefore include a higher degree of randomness. They are akin to carrying out 100 10 year re-shuffled simulations. So we were happy with their results – and the simple comparison of Random 1 and 2 showed they were performing correctly. If these runs had involved a single shuffling at the beginning of the 1000 year then we would expect that each run would show much greater difference in the erosion patterns seen.

P10 L8-9: if this issue is so important, it should be introduced as an objective of the paper. As written it appears like an additional side issue. The description of these jumbled runs should be added in the method section and removed from the discussion.

This has been changed. Moved to methods/results and removed from discussion.

P10 L5-10: I fully agree with the authors that it is a major limitation to this study. Thus I recommend the authors to try to assess how m and Ks variations could impact the sensitivity analysis as it will help to generalize their findings.

We believe that Reviewer 1 is referring to Page 12 here. The CAESAR-Lisflood model does not presently allow for a variable K value. This study looks solely at the influence of the rainfall input resolution on the modelled sediment yields and geomorphology of the catchment. By allowing for spatial variation of m and K as suggested, this would add a further variable from the standard approach of using a global value for these parameters. It is our belief that, although it would be an interesting avenue of investigation (indeed, similar work by one of the authors has recently been published in this area), it would only confuse the purpose of the investigation in this manuscript.

P10 L10: Why 20 different records? Does this number has an impact on the range covered in Figure 7 (i.e. from -7 to 2,5% for X-axis and from -15 to 60% for Y-axis)?

Twenty was chose as there are 20 rain cells in this domain -and this showed a clear pattern, as would have with ten. Running for 100 tests might extend the range covered in Figure 7, but not the step difference between the temporal resolutions which was the point of the Jumble tests.

P18 Table 3: Evaporation was set to 0. How does it impact the conclusions? K is missing in this table.

K is not a variable parameter in C-L - Evaporation was set to 0, yet this was constant throughout the tests. It will make a small difference, as would including the vegetation parameters or a bedrock layer making the region transport limited. This needs to be viewed as a conceptual sensitivity test into rainfall resolution alone. Varying other factors is beyond the scope of this study.

P27 Figure 6: Total rainfall : The authors should specify over how many years.

Over the available NIMROD record (ten years).

P28 Figure 7: Why was this analysis done on the upper Swale only? The complete basin is characterised by more rainfall cells and would have probably exhibited more variations in the random redistribution of rainfall (see author's comment in section 4.2, line 5). I find this figure very interesting in addition to the results from Table 8 for example, as it clearly shows the impact of the jumbled runs. This sensitivity of the sediment yield to different spatial and temporal distribution of the rainfall raises again the question : would this sensitivity be the same if also m and K parameters were included in a wider jumbled run numerical analysis.

Computational efficiency was the main reason. This involved 160 tests - each taking roughly 8 hours for the upper Swale, and 2 - 3 days for the Complete Swale. Also consistency - the majority of the additional tests (1000 year, Random 1 and 2) were conducted using the Upper Swale. We have added a note to indicate that long model run times restricted how many simulations we could carry out.

Again, we feel the global values for m and K are justified for this study for reasons stated previously throughout the response. By varying the values in each cell, it would make it difficult to disentangle the influence of the rainfall resolution, and the influence of varying these values.

Technical correction:

P2 L15 : delete reference at the end of the sentence

Done

P3 L19-20 : can not find those three references in the reference list

Thank you – they had been manually added not in the referencing SW!

P3 L28 : Coulthard et al. (2013a)

this reference seems correct ?

P5 L4 : Add units for Qtot P6 L11-13 : n (Manning) is missing in the list

Added.

P7 L29: were then compared

Thank you.

P28 Figure 7: homogenize the colors for 6 hour (yellow and purple)

Changed.

---

## Author Comment (AC4) · 17 Jun 2016

Dear authors,

My apologies that it took a while to post this which is due to a misunderstanding. I think you have received two insightful reviews of the manuscript that you submitted. In my view the paper can be published in ESurf if these remarks (and the ones suggested by Declan Valters) are accounted for in a revised version. The main comment of ref. 1 relates to the potential interaction of soil properties and resolution. I most certainly agree with the fact that this is a valid point that should be dealt with in a revised manuscript. However, it is in my view not necessary to carry out a large number of additional simulations to investigate this as this would be a different research topic altogether. I do think though that you should discuss this issue (and its potential implications) in a revised version of the MS. You already addressed the issue along those lines in an author comment that can form the basis for the rewriting of this part of the MS

Reviewer #2 has concerns with the way you deal with increased temporal rainfall resolution. As erosion is indeed a non-linear function of discharge/rainfall intensity nonlinear effects are indeed to be expected. Reading your MS I do feel these remarks are important but may partially be caused by a misunderstanding of the procedure you used and of the aims of your study, which focuses on spatial patterns rather than total erosion amounts. Please clarify this in a revised manuscript: I am sure that this will also be of great help to other researchers who want to better understand your research.

Kind regards,

Gerard Govers, Associate Editor

*Thank you for the comments and guidance. We apologise for the delay in the response – the questions posed and additional work carried out proved quite complex to blend with the original manuscript resulting in many changes – that we think improve the paper considerably.*

*We have added text and references to acknowledge the points made by Reviewer 1 – which are important, but we feel beyond the scope of this paper. In line with Reviewer 1 we have made some major changes removing the section on basin size comparison and shifting the orographic effects section to the methods/results rather than being in the discussion.*

*For Reviewer 2 we carried out a number of additional simulations to address the question as to whether or not any differences in sediment yield could be calibrated or adjusted for. This was indeed possible, but showed interesting, and important spatial changes in erosion and deposition patterns that were due to this adjustment/calibration process. We felt these were both important and built upon the aims and objectives of the paper. All changes are described fully in the responses to the reviewers and highlighted in the tracked changes MS submitted.*

*Best wishes, Tom Coulthard and Chris Skinner*

---

## Editor Decision (ED1)

Dear authors,

My apologies that it took a while to post this which is due to a misunderstanding. I think you have received two insightful reviews of the manuscript that you submitted. In my view the paper can be published in ESurf if these remarks (and the ones suggested by Declan Valters) are accounted for in a revised version. The main comment of ref. 1 relates to the potential interaction of soil properties and resolution. I most certainly agree with the fact that this is a valid point that should be dealt with in a revised manuscript. However, it is in my view not necessary to carry out a large number of additional simulations to investigate this as this would be a different research topic altogether. I do think though that you should discuss this issue (and its potential implications) in a revised version of the MS. You already addressed the issue along those lines in an author comment that can form the basis for the rewriting of this part of the MS

Reviewer #2 has concerns with the way you deal with increased temporal rainfall resolution. As erosion is indeed a non-linear function of discharge/rainfall intensity nonlinear effects are indeed to be expected. Reading your MS I do feel these remarks are important but may partially be caused by a misunderstanding of the procedure you used and of the aims of your study, which focuses on spatial patterns rather than total erosion amounts. Please clarify this in a revised manuscript: I am sure that this will also be of great help to other researchers who want to better understand your research.

Kind regards,

Gerard Govers, Associate Editor

*Thank you for the comments and guidance. We apologise for the delay in the response – the questions posed and additional work carried out proved quite complex to blend with the original manuscript resulting in many changes – that we think improve the paper considerably.*

*We have added text and references to acknowledge the points made by Reviewer 1 – which are important, but we feel beyond the scope of this paper. In line with Reviewer 1 we have made some major changes removing the section on basin size comparison and shifting the orographic effects section to the methods/results rather than being in the discussion.*

*For Reviewer 2 we carried out a number of additional simulations to address the question as to whether or not any differences in sediment yield could be calibrated or adjusted for. This was indeed possible, but   showed interesting, and important spatial changes in erosion and deposition patterns that were due to this adjustment/calibration process. We felt these were both important and built upon the aims and objectives of the paper. All changes are described fully in the responses to the reviewers and highlighted in the tracked changes MS submitted.*

*Best wishes, Tom Coulthard and Chris Skinner*

**The sensitivity of landscape evolution models to spatial and temporal rainfall resolution:**

**Reviewer 1 Comments**

This paper deals with a very interesting and relevant question for the scientific community working on sediment transfers in mesoscale river basins: how do the spatial and temporal resolution of the meteorological forcing impact modelled sediment
5   yields? While this issue has already been addressed from a purely hydrological standpoint, it remains understudied in modelling approach dealing with landscape evolution and soil erosion. However it seems to me that the conclusions raised by the authors are not supported by enough simulations. My main concern is about the potential effect of changes in soil hydrological properties (spatially and temporally) as the spatiotemporal resolution of rainfall is changed. This is not at all considered by the authors in their simulations while they recognise at the end of the discussion that it may change
10   considerably the sensitivity of landscape evolution models to rainfall resolution. As hydrological properties might be scale-dependant, changing only the spatiotemporal resolution of rainfall between runs without considering potential scale interactions between rainfall and soil behaviour may lead to erroneous conclusions on the sensitivity of landscape models. I know that adding runs in which the soil properties are randomly changed (m and K parameters) will need considerable additional computation time but the conclusions of the paper would be more supported and strengthened.

*We would like to thank the reviewer for their comments and thorough review. Aside from typo's and other minor points/clarifications, the main point the reviewer asks us to address is the interaction with soil properties and the balance between precipitation (P) and infiltration (I).*

20   *We agree completely with the reviewer that soil and land use properties might influence our results. However, the focus of this study is to examine just the impact of spatial and temporal rainfall resolution. In our parameterisation, hydrological factors that will change spatially are deliberately treated globally so we can look solely at the role of rainfall resolution. The experimental set up (e.g. having different hydrological areas defined by the rainfall grid resolution) is contingent upon the deliberately limited research questions we are asking – and to look at both soil properties and rainfall resolution would, we
25   suggest, require a completely different model set up.*

*We believe that it is important to consider that basin hydrology – both in terms of soil properties – and the driving precipitation – is often dealt with incredibly simplistically in LEM's, if at all! Therefore, our motivation is to explore not just the sensitivity to resolution – but to show the difference between having no representation and some representation of a
30   distributed hydrology in LEM's.*

*Since this paper was first submitted – we have also submitted (and now published in early view) an article that takes a tightly constrained look at the impact of spatial changes in the TOPMODEL m value on the geomorphic outputs over longer time scales (Coulthard & Van De Wiel, 2016). There is certainly a place for a study looking at both together. This could be*

*looking at a combination of the two approaches opens up the CAESAR-Lisflood model to a framework of modelling using Hydrological Response Units (HRU), a common approach in semi-distributed hydrological models, such as Dynamic-TOPMODEL and SWAT. This allows rainfall, land cover and soil properties to be represented at higher resolution than a global lumped estimate, but divided into broadly hydrologically homogenous regions.*

*Whilst we have not carried out any additional research to answer the points raised by the reviewer above and below, we certainly accept their validity – and have added a section to the discussion/limitations section*

*Coulthard, T. J., & Van De Wiel, M. J. (2016). Modelling long term basin scale sediment connectivity, driven by spatial land*
10 *use changes. Geomorphology. http://doi.org/10.1016/j.geomorph.2016.05.027*

Concerning the structure of the manuscript, the result section is very short and could be expanded, particularly if additional simulations are presented. The discussion section is rather heterogeneous in answering the 3 research questions written in the introduction. Section 4.2, addressing question 2, is very short and does not fully address it, as the authors recognise that more
15 simulations would be required. Also Section 4.3, addressing question 3, is not supported by the data (no reference to them). I would suggest focusing the results and the discussion on question 1.

We have changed the research questions – and dropped number 2.

20 If this question is fully addressed according to the above mentioned issues dealing with the hydrological basin properties, this would represent a substantial contribution to earth surface mass transfers. For those reasons, I do not recommend acceptance of the manuscript in its present form.
Specific remarks:
P2 L31-33: "Improved model performance" is not only "tempered by increased uncertainty surrounding precipitation data",
25 but by the uncertainty in the budget of precipitation (P) versus infiltration (I) or storage in the soil.
P3 L8-9: I fully agree with the authors here. This also refers to the ability to simulate correctly P-I budgets, spatially and temporally. As argued in the general comments, the authors should try to address that issue in their numerical sensitivity analysis.

30 See first comment. There is of course a need to look at the sensitivity of models to both spatial and temporal changes in precipitation AND land use – but in this paper we have focused on just one. This is to (a) make the experimental design simpler and (b) because spatial changes in land cover is really a different research/science question that we have answered in a subsequent paper.

P4 L30: I found the description of the model spatial discretisation quite confusing (not sure to have completely understood yet), mentioning here "area lumped parameters" and later (P5 L18) "grid cell size Dx" without giving any typical size for Dx. Is it the DEM resolution (50m) mentioned

5  Changed to say 50m.

 P7 L31? Could the authors try to be more specific on spatial discretisation and if possible limit the reference to previous papers to very specific model details that are not essential for that study?

10  There are no references in this paragraph, the three before or the two

P6 L1-3: As far as I understand, the hydrological model is adapted to the rainfall grid. Thus I agree with the authors that it enables having different levels of storage and runoff in each cell, but only due to rainfall variations. Varying also m and K would also create variations (i.e. P-I budget), but the authors kept constant those parameters (P8 L3).

This is true - but the cells do not cover hydrologically homogenous areas and values of m and k would be difficult to determine. m and k are not scale dependent, therefore the use of global values here is justified, yet we acknowledge that adjusting these values to local conditions will change the outputs - this is beyond the scope of this study (which is motivated only by rainfall resolution).

P7 L29: How can the authors be confident in their conclusions with only 2 additional long term random simulations?

In each random scenario the rainfall distribution was reshuffled every ten years, producing a significant element of randomness to the full 1000 year record simulated. However, both these two random runs produced similar results to each 25  other. Additional random scenarios could be run, yet this is computationally expensive and would yield similar results. The random runs were motivated by wanting to disrupt the spatial pattern in the ten year record, which is repeated 100 times in the long term runs - in reality, we could have been confident with just one test as this achieved this, but the only very minor variations between the two reinforce this further.

30  We can see that this may not be clear from the text – so this has been clarified in the methods section. Part of this is trying to explain how methods evolved in response to results – but within the methods section!

P8 L1 : Which initial grainsize distribution was used to run the 30 year model used as an initial condition? Which grain sizes are given in Table 3?

The grainsizes in Table 3 are those used to initialise the model with a global distribution, which was then spun up using a thirty year simulation. The spun-up grain size distribution was used for the tests. Text has been added to make this much clearer.

P8 L16: "considerable differences": this is not new and references should be given to situate these data.

Yes - this well established and long been understood. Line 16 has been altered in the manuscript to make this clearer.

10   P8 L21-23: I agree that these changes are minor. Could they be significantly different if m and K parameters were also randomly changed from one run to another?

Possibly, but as previously mentioned that is not the aim of this study.

15    P8 L28: "also drains an additional tributary". This may be critical. What is the drainage area of this tributary? How are the authors confident with that comparison between model and measurements?

The purpose of the hydrology tests was not to assess the model's performance and skill at estimating the hydrological conditions, but to demonstrate the effects on the hydrology when driving the model with different resolutions of input data.
20   We believe that the data presented do this effectively.

[Figure]

The image above shows the full extent of the Swale catchment we have used, plus the location of the Catterick Bridge gauge station and the missing drainage network. The system is predominantly Gilling Beck , which becomes Skeeby Beck before entering the River Swale. The information for the Catterick Bridge Gauge station (http://nrfa.ceh.ac.uk/data/station/info/27090) states it has a draingage area of 499.4km$^2$ - of which, the Complete Swale DEM is 415km$^2$ - so the missing area covers 17% of the drainage area. It's overall contribution to the discharge of the River Swale is not known.

P9 L11 : Very little difference is observed between random 1 and 2. This relates to a previous comment. Are 2 random simulations enough? Why not having done more, as presented later with the jumbled runs (P10 L9) for answering another question. Could these differences be more important if additional random simulations were performed? Otherwise this result (little difference) contradicts somehow with the results in Figure 7 showing a great dependence of the sediment yield to the rainfall allocation. Overall, it seems to me that the authors tried to address too many questions in the paper without running enough simulations to address each of them.

The jumble runs involve the shuffling of the rainfall distribution just the once. Random 1 and 2 each involve 100 reshuffles of the rainfall distribution and therefore include a higher degree of randomness. They are akin to carrying out 100 10 year re-shuffled simulations. So we were happy with their results – and the simple comparison of Random 1 and 2 showed they were performing correctly. If these runs had involved a single shuffling at the beginning of the 1000 year then we would expect that each run would show much greater difference in the erosion patterns seen.

P10 L8-9: if this issue is so important, it should be introduced as an objective of the paper. As written it appears like an additional side issue. The description of these jumbled runs should be added in the method section and removed from the discussion.

This has been changed. Moved to methods/results and removed from discussion.

P10 L5-10: I fully agree with the authors that it is a major limitation to this study. Thus I recommend the authors to try to assess how m and Ks variations could impact the sensitivity analysis as it will help to generalize their findings.

We believe that Reviewer 1 is referring to Page 12 here. The CAESAR-Lisflood model does not presently allow for a variable K value. This study looks solely at the influence of the rainfall input resolution on the modelled sediment yields and geomorphology of the catchment. By allowing for spatial variation of m and K as suggested, this would add a further variable from the standard approach of using a global value for these parameters. It is our belief that, although it would be an interesting avenue of investigation (indeed, similar work by one of the authors has recently been published in this area), it would only confuse the purpose of the investigation in this manuscript.

P10 L10: Why 20 different records? Does this number has an impact on the range covered in Figure 7 (i.e. from -7 to 2,5% for X-axis and from -15 to 60% for Y-axis)?

Twenty was chose as there are 20 rain cells in this domain -and this showed a clear pattern, as would have with ten. Running for 100 tests might extend the range covered in Figure 7, but not the step difference between the temporal resolutions which was the point of the Jumble tests.

P18 Table 3: Evaporation was set to 0. How does it impact the conclusions? K is missing in this table.

K is not a variable parameter in C-L - Evaporation was set to 0, yet this was constant throughout the tests. It will make a small difference, as would including the vegetation parameters or a bedrock layer making the region transport limited. This needs to be viewed as a conceptual sensitivity test into rainfall resolution alone. Varying other factors is beyond the scope of this study.

P27 Figure 6: Total rainfall : The authors should specify over how many years.

Over the available NIMROD record (ten years).

P28 Figure 7: Why was this analysis done on the upper Swale only? The complete basin is characterised by more rainfall cells and would have probably exhibited more variations in the random redistribution of rainfall (see author's comment in section 4.2, line 5). I find this figure very interesting in addition to the results from Table 8 for example, as it clearly shows the impact of the jumbled runs. This sensitivity of the sediment yield to different spatial and temporal distribution of the rainfall raises again the question : would this sensitivity be the same if also m and K parameters were included in a wider jumbled run numerical analysis.

Computational efficiency was the main reason. This involved 160 tests - each taking roughly 8 hours for the upper Swale, and 2 - 3 days for the Complete Swale. Also consistency - the majority of the additional tests (1000 year, Random 1 and 2) were conducted using the Upper Swale. We have added a note to indicate that long model run times restricted how many simulations we could carry out.

Again, we feel the global values for m and K are justified for this study for reasons stated previously throughout the response. By varying the values in each cell, it would make it difficult to disentangle the influence of the rainfall resolution, and the influence of varying these values.

Technical correction:

P2 L15 : delete reference at the end of the sentence

Done

P3 L19-20 : can not find those three references in the reference list

Thank you – they had been manually added not in the referencing SW!

P3 L28 : Coulthard et al. (2013a)

this reference seems correct ?

P5 L4 : Add units for Qtot P6 L11-13 : n (Manning) is missing in the list

Added.

P7 L29: were then compared

Thank you.

P28 Figure 7: homogenize the colors for 6 hour (yellow and purple)

Changed.

**The sensitivity of landscape evolution models to spatial and temporal rainfall resolution:**
**Reviewer 2 Comments**

*We have made considerable changes to the MS based on this review – both including a number of additional references and*
5  *in a series of new simulations to address the final point made by the reviewer. These are detailed below and in the revised*
*MS.*

This paper examines the effect of temporal and spatial resolution has the erosion and landform evolution predictions of a
LEM. The broad conclusions of the paper are a worthwhile contribution but the discussion misses some important points and
10  previous work, and misrepresents previous work by other authors.

*We would like to thank the reviewer for their comments and to apologise for completely missing developments made by*
*previous authors. Some of the comments by the reviewer refer to unpublished research examining the role of temporal*
*rainfall resolution – and make complete sense as does the thought experiment outlined in the review. However, it is difficult*
15  *to reference unpublished findings, but we have looked in some detail at the SIBERIA literature, finding a relevant section in*
*a user manual and used this accordingly in the revised MS. The calibration process outlined in the Willgoose and Riley*
*(1998) paper makes no direct reference to rainfall resolution – but having read the reviewers comments – and re-read the*
*paper sections it is clear that this is part of the calibration process. We have added sections and re-worked parts of the*
*paper to clearly acknowledge this. Our findings (with regard to temporal rainfall precipitation) certainly agree with those*
20  *mentioned above – and this is duly noted.*

*We considered removing the temporal component of the model comparison and focusing on the spatial in the revised MS, but*
*thought that our experiments still contained an important contribution as it looked at how the relationship changed through*
*different resolutions as well as over different basins. Additionally it also allowed the combination of spatial and temporal*
25  *rainfall resolution to be examined. Therefore, it represents a systematic investigation into rainfall spatial and temporal*
*resolution.*

First looking at the question of spatial resolution. It's rather hard to judge the results without some idea of what the spatial
pattern of rainfall is in the 10 year record and how persistent this pattern is over the 10 year period. A couple of thought
30  experiments will clarify my concerns.
1. Imagine now that the pattern remains exactly the same over the 10 year period (i.e. the amount of rainfall over the
catchment changes from year to year, but the pattern of this rainfall is exactly the same form year to year). Then the random
redistribution of rainfall in space will be completely invalid since what is required a random resampling of the rainfall in
each year. This is an extreme case of orographic rainfall.

2. Imagine now that the pattern is completely uncorrelated from year to year and from 5km pixel to 5km pixel. In this case the random redistribution will be OK and any changes will simply result in random noise in the erosion and landform results. The authors have failed to justify that the differences they observe are anything other than random effects.

*We have to be careful to consider that in reality rainfall is not random. It does have patterns (spatial and temporal) – and some of these temporal patterns should be retained otherwise the resampled/ chopped record is meaningless. Therefore, we have not randomly re-sampled during the year – as the rainfall is made of 'events' – here largely associated with frontal rainfall. It would be unrealistic to distintangle these events – so you would (for example) have one pixel of heavy rain pop up*
10 *in the middle of a dry spell. We have done this to a degree by spatially 'mixing' every 10 years – but the mixed pixels are still in temporal sync with each other. This could be broken down into annual mixing – but over a 1000 year simulation would that really give a different solution from our one? As figure 4 shows, there is relatively little difference between two of our random 1000 year simulations.*

15 *What we have done by spatially mixing the rain cells every 10 years in a 1000 year run, is to show the aggregate of 100 mixed up, 10 year simulations (its an easier way than showing an average if you like). By having different mixed up runs that give very similar results spatially and in bulk yields (Figure 4) – yet clearly different from the non mixed up results (figure 5) - we show that we can remove any spatial bias in the patterns of rainfall we are using in these 1000 year simulations. This means that we can compare 5km spatially distributed (randomly mixed spatially every 10 years) to lumped rainfall*
20 *simulations over the same period.*

*A neater solution to this issue would be to use a synthetic rainfall generator that also simulates spatial patterns of rainfall. These exist, though are relatively new and less tested than non spatial rainfall generation methods (e.g. Peleg & Morin 2014). Here, this would significantly expand the work required, scope and aims of the paper (in effect, it is another paper).*
25 *Peleg, N. & Morin, E., 2014. Stochastic convective rain-field simulation using a high-resolution synoptically conditioned weather generator (HiReS-WG). Water Resources Research, 50(3), pp.2124–2139. Available at: http://doi.wiley.com/10.1002/2013WR014836 [Accessed June 17, 2016].*

The way the question about time resolution is posed shows a misunderstanding of some of the solutions that other workers
30 have used to address the problems highlighted of differences in mean erosion rate observed by the authors. There is no question that high time resolution runoff series results in significantly increased in erosion rates. The reviewer has also seen this in his our erosion computations and the 100% increase from daily to 0.25 hour accords with our own, unpublished, experience. This is because of the nonlinear dependence of the erosion time series on the runoff time series. A simple first order second moment analysis of the erosion time series shows this.

Consider an erosion equation that is dependent on the square of discharge (approximately the dependency of Einstein-Brown sediment transport equation)

$$E = bQ^2 \quad (1)$$

If Q is now a random value with mean Q* and variance SQ. A second order first moment analysis of this equation yields

$$E = b(Q^{*2} + SQ) \quad (2)$$

So that the erosion is higher than that where there is no variation in Q by a factor

$$(1 + SQ/Q^{*2}) \quad (3)$$

This analysis shows that the erosion rate when you allow for randomness versus where you average out the variability will always be higher and the percentage increase is a function of the coefficient if variation of the runoff series.

My own observation is that this factor can easily by a factor 2 going from a daily runoff series to a 15 minute runoff series for a small catchment (i.e. the erosion will increase by 100%). The appearance of variance in equation (2) comes solely from the square dependence in equation (1). If equation (1) were a power of 1 (i.e. linear) then the variance term does not appear and the sub-daily variability would have no impact on the mean erosion rate.

*This is a really interesting way of breaking down the issue for temporal rainfall – in our representation, erosion (with the addition of various parameters) is roughly the square of the velocity – so a similar relationship. We would have liked to include a similar breakdown in the revised MS – but would not want to make this look like our thoughts (and we cannot readily cite reviews). Hopefully, the quote from the SIBERIA manual we have included covers part of this (certainly the last para above).*

Finally, the authors quote Hancock papers (2000,2002,2010) as examples where the long time resolution of the timesteps in the landform evolution model will yield significant underestimates of the erosion. This assertion is categorically incorrect and reflects a lack of understanding of how the model parameters were developed for these papers. I'm surprised at this because the first author has been collaborating for some time with Hancock. The parameters used in the Hancock papers are based on a calibration procedure described in Willgoose and Riley (1993,1998) Willgoose, G. R., and S. J. Riley (1993), Application of a catchment evolution model to the prediction of long-term erosion on the spoil heap at Ranger Uranium MineRep. Open File Report 107, The Office of the Supervising Scientist, Jabiru. Willgoose, G. R., and S. J. Riley (1998), An assessment of the long-term erosional stability of a proposed mine rehabilitation, Earth Surface Processes and Landforms, 23, 237-259.

In brief this process was 1. A conceptual rainfall-runoff model (with much the same capability as LISFLOOD) was calibrated to rainfall-runoff-erosion plot studies at the time and space resolution of the data (minutes and 100 sq metres) 2. A multiple regression was developed between sediment load, discharge and slope from the plot studies. 3. The rainfall-runoff model was then scaled up to the landform using a low resolution DEM of the site (about 1000 nodes) and 30 years of

pluviograph data at 15 minute resolution was used to generate a 30 year runoff time series. 4. This 15 minute resolution time series was then used to generate a 15 minute sediment transport series using the regression. 5. This 15 minute erosion series was then lumped up to the annual level and "effective" parameters where developed that gave the same average and area and slope dependence at the yearly time step as the 15 minutes erosion time series. These are the parameters that are used in the annual time steps.

Now there is no doubt this was an extremely compute intensive task. In 1992 when this work was done it took about 4 weeks of CPU time on a high end workstation to generate the time series in step 3. This calibration has been used as the basis for other sites studied by Hancock.

The key difference between what was done by Willgoose and Riley (1998) (hereafter W&R) and in this paper is that the authors have explicitly included the randomness of the hydrology timeseries within the LEM, while in W&R this has been averaged out in the derivation of the effective parameters.

Finally on bottom of p10 and top of p11 the author contemplates whether there is a "compensatory factor or exponent". Indeed this is what the "effective parameters" in the approach of W&R do.

So in conclusion if we go to the plots of changes when using different averaging periods, the lower erosion rate observed by the author for low resolution rainfall is to be expected. But this can be adjusted by the use of "effective parameters" as done in W&R.

The more interesting question, but unfortunately not addressed by the authors, is if the average erosion rate for all the different time resolutions were adjusted to give the same annual erosion are the landforms generated significantly different (i.e. does the higher rainfall resolution and explicit modelling of runoff events lead to fundamental differences beyond a general change in the calculated mean erosion rate).

This *is* a really interesting question – and we are grateful for the reviewer for suggesting this. In the revised MS we have now done just this – to adjust model runs (via a compensation factor in the sediment transport law) so very similar sediment yields (erosion rates) are generated over 1000 year simulations. Rather than try and tune all our simulations to the same erosion rate (and therefore to reduce the number of simulations needed) we adjusted some simulations (e.g. 15 min lumped) too match existing results (e.g. 24 hour lumped). This required an additional 30 simulations – each taking 4-8 weeks. This generated some really interesting findings – and as the reviewer suggested – does lead to considerable differences in the spatial patterns of erosion and deposition found within the basin.

These simulations and research, have resulted (in the paper) in additional sections in the methods, results, discussion and conclusions – and we think they significantly enhance the paper and its findings.

[revised manuscript text omitted]